# Recruitment of plasma cells from IL-21-dependent and IL-21-independent immune reactions to the bone marrow

Marta Ferreira-Gomes [1,10], Yidan Chen [1,2,10], Pawel Durek[1,10], Hector Rincon-Arevalo [1,2,3,4], Frederik Heinrich [1], Laura Bauer [5], Franziska Szelinski [1,2], Gabriela Maria Guerra[1], Ana-Luisa Stefanski [1,2], Antonia Niedobitek[1], Annika Wiedemann[1,2], Marina Bondareva[1], Jacob Ritter [1,2], Katrin Lehmann[1], Sebastian Hardt[6], Christian Hipfl[6], Sascha Hein [7], Eberhard Hildt [7], Mareen Matz[8], Henrik E. Mei [1], Qingyu Cheng[1,2], Van Duc Dang[1,2], Mario Witkowski [1,9], Andreia C. Lino [1,2], Andrey Kruglov[1], Fritz Melchers[1], Carsten Perka[6], Eva V. Schrezenmeier[1,4,8], Andreas Hutloff [5], Andreas Radbruch [1,2,11], Thomas Dörner[1,2,11] & Mir-Farzin Mashreghi [1,11] ✉

Bone marrow plasma cells (BMPC) are the correlate of humoral immunity, consistently releasing antibodies into the bloodstream. It remains unclear if BMPC reflect different activation environments or maturation of their precursors. Here we define human BMPC heterogeneity and track the recruitment of antibody-secreting cells (ASC) from SARS-CoV-2 vaccine immune reactions to the bone marrow (BM). Trajectories based on single-cell transcriptomes and repertoires of peripheral and BM ASC reveal sequential colonisation of BMPC compartments. In activated B cells, IL-21 suppresses CD19 expression, indicating that CD19$^{low}$-BMPC are derived from follicular, while CD19$^{high}$-BMPC originate from extrafollicular immune reactions. In primary immune reactions, both CD19$^{low}$- and CD19$^{high}$-BMPC compartments are populated. In secondary immune reactions, most BMPC are recruited to CD19$^{high}$-BMPC compartments, reflecting their origin from extrafollicular reactivations of memory B cells. A pattern also observable in vaccinated-convalescent individuals and upon diphtheria/tetanus/pertussis recall-vaccination. Thus, BMPC diversity reflects the evolution of a given humoral immune response.

Long-lived plasma cells (PC) are rare cells that reside and rest in dedicated survival niches of the bone marrow (BM), which continuously secrete antibodies against previously encountered pathogens[1]. These cells have been considered to originate mostly from T cell-dependent B-cell activation in secondary lymphoid organs. The concept of long-lasting humoral immunity provided by long-lived (memory) plasma cells[2,3] has gained acceptance and attention during the COVID-19 pandemic, when the presence of spike-specific memory plasma cells in the BM was considered a correlate of long-lasting

protection against severe disease and death[4,5]. This protection, conferred by stable systemic antibody titres, can last for decades, if not a lifetime[6]. Despite their relevance, the recruitment of plasma cells from secondary lymphoid organs into the BM and their establishment there as long-lived plasma cells is not well understood. Plasmablasts generated in primary and secondary immune reactions might differ in their competence to enter the BM and survive there as long-lived plasma cells[1]. BM plasma cells (BMPC) are heterogeneous with respect to expression of e.g., CD19[7,8], CD38[7], PD-1[8], CD39 and CD326[9], but

whether this reflects different environments encountered during their generation, their maturation in the BM or different qualities as BMPC, e.g., life-spans, has remained unclear. It also is unclear whether BMPC are constantly recruited to the BM during an immune reaction or only at the end, when affinity maturation is finished[10–12].

Here, we address these conceptual questions by performing a global analysis of the transcriptional heterogeneity and antigen-receptor repertoire of human BMPC, and we follow the recruitment of plasma cells (PC) to the BM in primary and secondary immune reactions to SARS-CoV-2 vaccines. We find 10 "clans" of BMPC, compartments differing in and reflecting the instructive signals they received as activated B cells. By comparing the transcriptomic signatures of newly generated circulating antibody-secreting cells (ASC) from peripheral blood to the transcriptomes of established BMPC, we track the recruitment of PC to the bone marrow in the primary and secondary immune reaction. We identify interleukin-21 (IL-21) as the signal to downregulate expression of CD19 on activated B cells. CD19low PC are thus generated in IL-21-dependent follicular germinal centre reactions, and CD19high PC are generated in IL-21-independent extra-follicular (re)activations of (memory) B cells. In the immune reactions analysed here, PC generated in primary reactions are recruited to distinct CD19low and CD19high clans, while PC generated in secondary immune responses are nearly exclusively recruited to clan 0 (CD19high). Most ASC exiting the immune response early on express IgG1—an isotype induced by IL-21[13,14]. However, as the immune response progresses, ASC measured at later time points also express IgA1 and IgA2, indicative of TGF-β instruction at the time of activation and class switching[15]. ASC expressing IgG2, reflecting instruction by interferons[16], were not as frequently induced by the vaccines. Taken together, PC recruited to the bone marrow in secondary immune reactions are derived from non-follicular reactivation of memory B cells, while CD19low BMPC reflect the direct IL-21-dependent output of primary germinal centre reactions. Upon repeated vaccination, spike-specific BMPC, but also tetanus-specific BMPC are present in most clans, reflecting the evolution of the respective immune response, the continued recruitment of PC to the BM during the immune reaction, and their lasting maintenance there.

## Results

### Transcriptional and phenotypic heterogeneity of BMPC

Single-cell transcriptomes of 49,347 BMPC were obtained from eight patients who underwent hip joint-replacement surgery (Supplementary Table 1). BMPC were enriched as viable CD38high CD138high CD3−CD10−CD14− or CD38high CD27high CD3−CD14− cells (Supplementary Fig. 1a), incubated in addition with DNA-barcoded antibodies for Cellular Indexing of Transcriptomes and Epitopes by Sequencing (CITE-seq), and subjected to single cell transcriptome as well as full-length B-cell receptor (BCR) sequencing. Cells were clustered according to their transcriptomes and visualised by uniform manifold approximation and projection for dimension reduction (UMAP)[17]. From the 15 defined subpopulations, clusters 2, 3, and 10 were excluded from further consideration, as they contained low quality cells expressing 16.4% (median, MAD = 6.6%) mitochondrial genes and/or a significantly low number of transcripts (a median of 453, 842, and 662, respectively; MAD = 133, 266, 152) (Supplementary Fig. 1b). Clusters 7 and 14 represented CD20-expressing B cells and pre-B cells, respectively, and thus were not considered further as BMPC. The remaining 10 clusters (Fig. 1a), amounting to 38235 cells and present in all eight donors (Fig. 1b), were classified as BMPC according to their expression of the signature genes PRDM1, IRF4, XBP1, and SDC1 (Fig. 1c), and surface proteins, including CD27, CD38 and CD138 (Supplementary Fig. 1c, d). 38% of the BMPC analysed resided within cluster 0, with 34% of them expressing CD19 transcripts and being of CD19high phenotype (Fig. 1b–d, Supplementary Fig. 1e). Most of these cells expressed IgA1 and IgG1 antibodies (Fig. 1e, f). Clusters 1 and 4

contained predominantly CD19low cells, 93% and 84%, respectively, not expressing CD19 transcripts (Fig. 1c, d, Supplementary Fig. 1e). While cluster 1 was enriched for IgG1-expressing BMPC, cluster 4 contained mostly IgM and IgA-expressing BMPC (Fig. 1e, f). Cells from cluster 5 are a population of XBP1low IRF4high IgA1 BMPC, with 27% expressing CD19 transcripts. Cluster 4 is enriched in IgA2 and IgM BMPC expressing IgJ, CCR10 and ITGB7, indicative of a mucosal origin and TGF-β instruction[18] (Fig. 1c–f). Cluster 6 consists of BMPC expressing CD9 and IgG2, hallmarks of a type II interferon (IFN)-driven immune reaction (Fig. 1d–f). Cells expressing elevated levels of STAT1 and IFITM1 (Fig. 1d) were contained in cluster 8, i.e. BMPC generated in a type I IFN response. This observation is supported by gene set enrichment analysis (GSEA), where the relative expression of genes associated with defined pathways is shown in a density plot (Fig. 1g). Clusters 9 and 13 consist of BMPC expressing HLA-DR, a hallmark of recent generation from activated B cells[19] (Fig. 1d, Supplementary Fig. 1c, d). They represent proliferating plasmablasts (cluster 13) and newly generated, still HLA-DR+ plasma cells (cluster 9), with PD-1 signalling GSEA indicating their recent interaction with PD-1 expressing T cells (Fig. 1g). BMPC of these two clusters also show a GSEA characteristic of high translational activity and oxidative phosphorylation (Fig. 1g). Finally, clusters 11 and 12 are BMPC expressing heat shock genes and NR4A1 (Nur77), respectively, (Fig. 1d) indicative of cellular stress. Nur77 has been described as an antagonist of B-cell lymphoma (Bcl)-family members and inducer of apoptosis in myeloma cells[20], as well as being linked to B and T-cell self-reactivity[21,22].

Overall, the increased FAS (CD95) and/or lower BCL2 expression in CD19high clusters (Supplementary Fig. 1d, f) suggest that they might be transcriptionally more prone to succumb to extrinsic and intrinsic apoptosis inducers, while CD19low clusters may be more resilient. In support of this hypothesis, GSEA indicates that cells of clusters 1 and 6 show STAT3, STAT5, and TNF receptor family signal transduction signatures. They also express IL5R as well as the TNF-family receptors TNFRSF13B (TACI) and TNFRSF17 (BCMA) (Fig. 1g, Supplementary Fig. 1f), the latter ones protecting plasma cells from anabolic stress[23–25]. Clusters 1 and 6 are also enriched in plasma cells expressing genes associated with hypoxia, a condition known to favour plasma cell survival in BM niches[26]. Finally, at the border between cluster 0 and 1, there is an enrichment of cells expressing genes associated with glycolysis, a metabolic condition crucial for plasma cell survival in BM niches[27] (Fig. 1g). In summary, analysis of the phenotype and transcriptional profile of individual BMPC reveals a remarkable heterogeneity, even within the 10 clusters identified, which thus define clans of BMPC, rather than homogeneous clusters.

### BMPC clans express exclusive antibody repertoires

Consistent with their origin from distinct types of immune reactions, the various BMPC clans also expressed different repertoires of antigen receptors. Overall, the different BMPC clans were similar in diversity and we did not find significantly expanded clones (Fig. 2a). The repertoires of the different clans, defined by the germline sequences of heavy and light chains of the BCR and CDR3 similarity of their individual cells, did not overlap beyond what would be expected stochastically (Supplementary Fig. 2a). Also, pseudotime analysis did not identify clear relationships and developmental paths between the BMPC clans (Supplementary Fig. 2b). Mutations in both the framework and the CDR sequences were more frequent in clans 0, 8 and 11, i.e., the CD19high BMPC, as compared to the CD19low BMPC clans 1, 4 and 6 (Fig. 2b–d, Supplementary Fig. 2c). This is in line with our previous findings[28].

In order to locate antigen-specific cells in the various plasma cell clans, we analysed BMPC secreting antibodies specific for the receptor-binding domain (RBD) or full spike protein of SARS-CoV-2, or the tetanus toxoid (TT) C fragment. All BM samples were processed during the COVID-19 pandemic, with patients having either

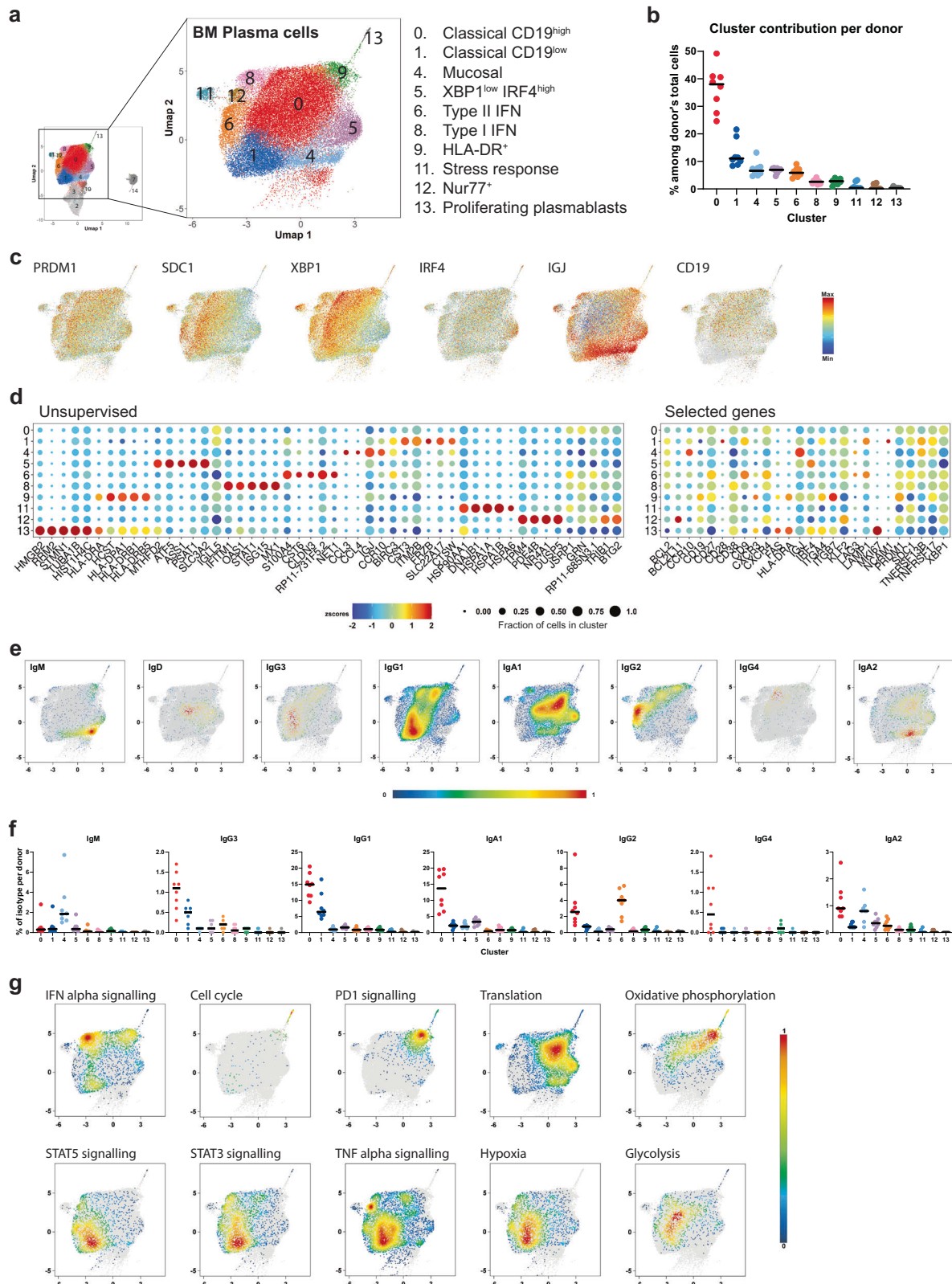

had contact with SARS-CoV-2 or been vaccinated between 56 and 312 days prior to BM sampling (Supplementary Table 1). In all cases where serum could be analysed (Supplementary Table 1), patients had RBD- and spike-specific serum antibodies, as well as TT-specific serum antibodies (Supplementary Fig. 2d). Since most BMPC do not express antigen-receptors on their cell surface[29,30], direct isolation of viable antigen-specific BMPC for single-cell sequencing was not possible. Rather, to obtain a pool of antigen-specific BCR sequences we sorted and sequenced individual RBD/spike- and TT-binding memory B cells and plasmablasts/plasma cells either from the peripheral blood or the BM of vaccinated individuals (Supplementary Fig. 1a, gating strategy 2). Based on the CDR3 regions in heavy and light chains, we designated as "public" RBD/spike- and TT-specific clones, those BMPC clones with more than 80% similarity (modelled

**Fig. 1 | Transcriptional and functional heterogeneity of BMPC. a** Bone marrow plasma cells (BMPC; CD138$^{high}$ CD38$^{high}$ or CD38$^{high}$ CD27$^{high}$) from 8 patients undergoing total hip replacement surgery were isolated and sorted by FACS for single-cell sequencing (gating strategy in Supplementary Fig. 1a). Amplified area: UMAP representation of remaining 38235 plasma cells after exclusion of contaminant and poor quality cells (see Supplementary Fig. 1b). Clusters of transcriptionally similar cells were identified using shared nearest neighbour (SNN) modularity optimisation. **b** Percentage of BMPC found in each cluster per donor's total cells analysed. Horizontal lines indicate the median. *n* = 8 independent donors. **c** UMAP representation of the expression levels of selected PC signature genes across BMPC clusters. **d** Bubble plots of expression levels of the top five marker genes for each cluster (left) and of additional selected genes (right). Colour scale shows the z scores of the average expression of a gene within the indicated cluster. Bubble sizes correspond to the fraction of cells expressing a defined gene within the indicated cluster. **e** Density plots of immunoglobulin isotype expression within the BMPC compartment. **f** Frequency of plasma cells expressing a defined immunoglobulin isotype, displayed according to the clusters in **a**, per donor's total cells where a full BCR could be identified. For better readability, each isotype was plotted separately, even though the percentages are related to the total BCRs from each donor. Horizontal lines indicate the median. *n* = 8 independent donors. **g** Density plots of BMPC significantly enriched in gene sets from different biological pathways as calculated by Gene Set Enrichment Analysis (GSEA). Source data are provided as a Source Data file.

optimum for CDR3 specificity[31]) shared between individuals. Of the 38235 BMPC analysed, 84 expressed spike-specific and 120 tetanus-specific public clonotypes (Fig. 2e). These public BMPC clones were present in several clans in all individuals analysed, with the exception of clan 12 and clans 11 and 13 for tetanus specificity (Fig. 2f, Supplementary Fig. 2e). Their percentage among total BMPC is highly individual-dependent being up to 2.03% in the CD19$^{high}$ compartment (0.05 to 0.32% for spike- and 0 to 2.03% for tetanus-specific clones) and up to 1.25% in the CD19$^{low}$ compartment (0.04 to 0.24% for spike- and 0 to 1.25% for tetanus-specific clones). Mutation rates in the framework were similar between spike- and tetanus-specific BMPC irrespective of the isotype they expressed, except for spike-specific IgA1 BMPC (Fig. 2g). These were significantly less mutated than tetanus-specific IgA1 BMPC. Mutation rates in the CDR regions show that tetanus-specific IgM, IgG2 and IgA2 BMPC accumulate significantly more mutations than their spike-specific counterparts (Fig. 2g). In summary, tetanus- and SARS-CoV-2 spike-specific BMPC expressed somatically mutated antibodies and were located in several clans in the donors analysed.

## BMPC heterogeneity reflects the different timings/stages of immune reactions

To follow the recruitment dynamics of PC to BM in repeated antigenic challenges of a given immune response, we compared single-cell transcriptomes of ASC exiting the immune reaction into the blood at various time points after primary and secondary vaccination to the transcriptomes of BMPC. This longitudinal clinical study included 36 healthy individuals who had received different vaccines (Supplementary Table 2), namely the mRNA vaccine Comirnaty (SARS-CoV-2 spike, BioNTech/Pfizer, BNT), the viral vector-based vaccine Vaxzevria (SARS-CoV-2 spike, Oxford/AstraZeneca, AZ), and/or the mixed protein vaccine Boostrix (diphtheria toxoid, tetanus toxoid and pertussis toxoid, GlaxoSmithKline, DTP). To obtain single-cell transcriptomes and BCR repertoires, viable CD3$^-$CD14$^-$CD27$^{high}$CD38$^{high}$ cells (Supplementary Fig. 3a) were isolated from peripheral blood 7 and 14 days after primary immunisation (BNT, AZ), 7 days and 7 months after secondary BNT immunisation, and 7 days after third BNT vaccination (Fig. 3a). Circulating ASC from 7 days after primary vaccination of convalescent COVID-19 individuals, as well as 7 days and 6 months after boost with DTP vaccine were also isolated. Altogether, we sequenced 55071 peripheral blood ASC, based on which a new UMAP was generated. A uniform distribution of the cells within the UMAP with respect to the vaccine protocol and the time post vaccination was observed (Supplementary Fig. 3b). Based on gene expression of *CD38, HLA-DMA* and *MKI67* (Fig. 3b) we classified the peripheral ASC into newly generated plasmablasts (*CD38$^{high}$HLA-DMA$^{high}$*), proliferating plasmablasts (*MKI67$^+$*), and plasma cells (*CD38$^{low}$HLA-DMA$^{low}$*)[19]. Cells from each stage preferentially expressed isotypes reflecting the cytokine milieu that they had been exposed to during their activation leading to antibody class switch recombination (Supplementary Fig. 3c). While the plasmablasts expressed mostly IgM, IgG1, IgG2, IgA1 and IgA2, the plasma cells expressed mainly IgA1.

Using the same pool of antigen-specific BCR sequences and criteria for similarity used for the analysis of spike-specific BMPC, we tracked public putatively spike-specific clonotypes among all peripheral ASC. Spanning the different time points analysed, a total of 749 ASC that expressed such clones were identified (Fig. 3c, d). 14 days after primary vaccination, when spike-specific serum antibodies became detectable (Supplementary Fig. 3d), the identified public ASC of the BNT and AZ vaccinees predominantly expressed IgG1 and IgA1 (Fig. 3d). However, spike-specific public ASC from AZ vaccinees showed higher somatic mutation rates. In both groups after the second immunisation (3 and 12 weeks after the first, respectively), almost all identified spike-specific clones were IgG1. Public ASC from the AZ/BNT vaccinees showed even higher mutation rates. At 7 months after BNT/BNT vaccination, public ASC expressed a variety of antibody classes and an increase in mutation rates. 7 days after a third dose of BNT, public spike-specific ASC expressed IgG1, IgG2 and IgA1, with increased mutation rates as compared to 14 days after primary and 7 days after secondary vaccination. Interestingly, convalescent individuals who had received just one dose of BNT 7 days earlier had predominantly IgG1 spike-specific BCRs, with similar mutation rates as individuals 7 days after a third BNT vaccination (Fig. 3d). The integration of both BMPC and blood ASC data sets (batch corrected for sequencing depth, effects among individuals and origin of the cells) into a new UMAP projection show that BMPC cluster with blood ASC (Supplementary Fig. 3e), an exception being the plasmablasts found in the BM (cluster 13), which cluster together with blood proliferating plasmablast (MKI67+, Fig. 3b).

We then defined gene signatures characteristic for the circulating ASC at the different time points (Supplementary Data 1), and from the public antigen-specific ASC (Supplementary Data 1), and projected them onto the transcriptomes of the BMPC by GSEA (Fig. 3e, Supplementary Fig. 4a–d). The gene signatures obtained from each individual vaccinee at a given time point after vaccination showed only marginal inter-individual variation (Supplementary Fig. 5a). 7 and 14 days following primary immunisation with BNT or AZ vaccines, the signatures of circulating ASC marked clans 0, 1 and 4 of BMPC, both in terms of frequency and enrichment score (Fig. 3e, Supplementary Fig. 4a, b), with BNT delegating more PC into clans 1 and 4 than AZ, and AZ more into clan 9 than BNT. This difference is likely due to the fact that AZ also elicits an immune response against its adenovirus components (Supplementary Fig. 5b), as well as T-REx HEK293 proteins from virus production[32]. The resemblance of ASC to BMPC clan 1 and particularly clan 4 also held true 7 days following secondary vaccination with BNT (when the primary dose was given 28-35 days previously). In stark contrast, those primed with AZ, or infected with SARS-CoV-2 and boosted later with BNT, developed circulating ASC resembling BMPC of clans 0, 9, and 13. Moreover, clans 0, 9, and 13 expressed the signatures of circulating ASC egressing into the blood 7 days after vaccination with DTP – a boost to already established long-term memory. In those vaccinated with a third dose of BNT, 7 days later, the circulating ASC also shared signature gene expression with clan 0 of BMPC, but also clans 4 and 5, i.e. IgA1 and IgA2 expressing CD19$^{low}$ BMPC.

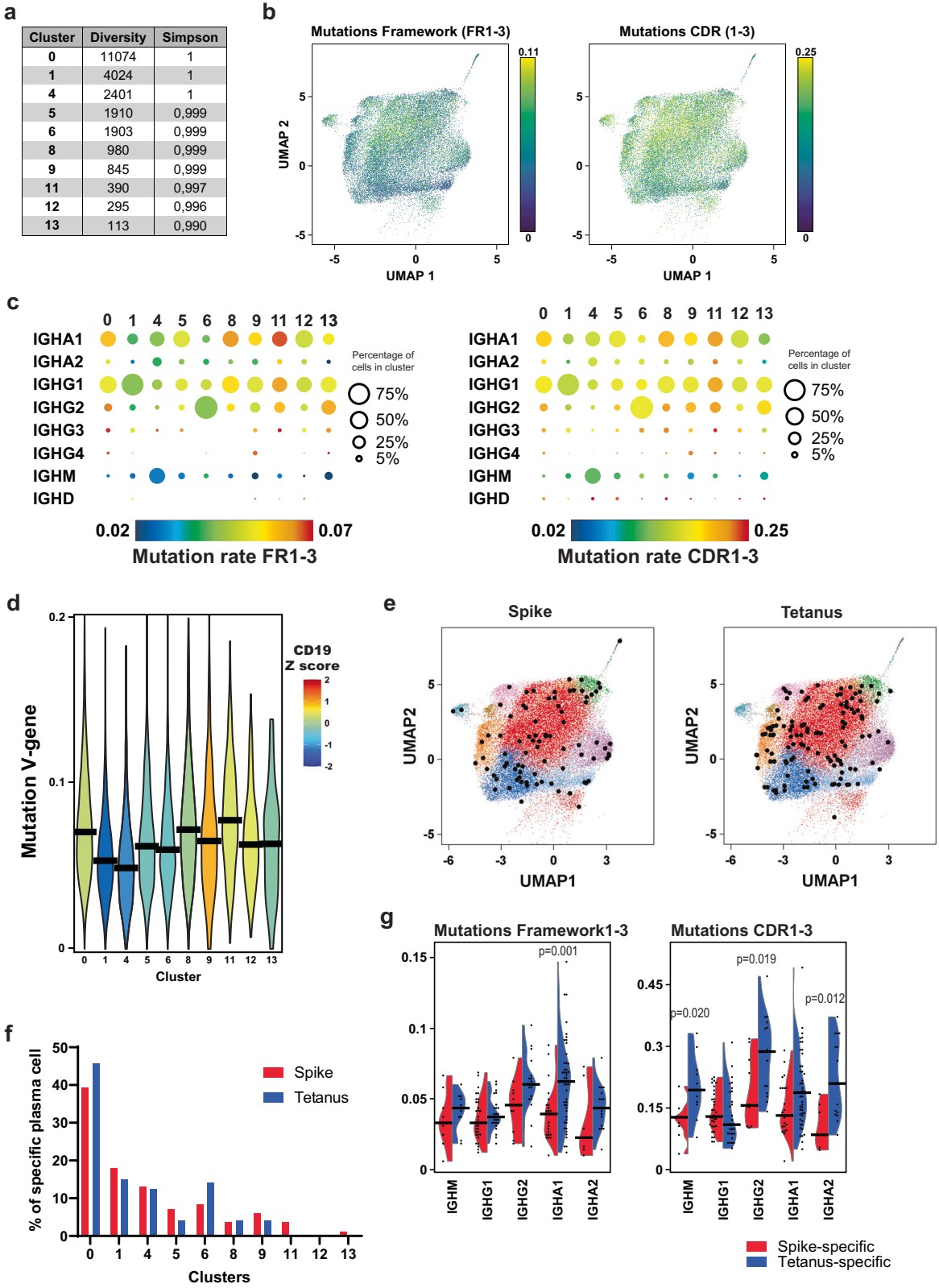

In order to obtain gene signatures from prolonged germinal centre output, we also analysed the circulating ASC 6/7 months after vaccination in individuals immunised twice with BNT or boosted with DTP[33]. At this time, we were still able to detect newly generated spike-specific ASC (Fig. 3d). Regarding ASC from BNT vaccinees, their signatures were mainly shared with BMPC of clans 0, 1, 5, 6 and 9 (Fig. 3e, Supplementary Fig. 4a, b). 6 months after DTP boost,

circulating ASC resembled the ones obtained 7 months after BNT/ BNT, with the exception of clan 1. This showed that while BNT prolonged response gives rise to CD19^high but also CD19^low BMPC, the prolonged response to a bona-fide DTP recall response gives rise predominantly to CD19^high BMPC. In summary, a prominent recruitment of PC to CD19^low clans is only observed upon primary and closely-followed secondary vaccination with

**Fig. 2 | BMPC are clonally diverse and present different mutation rates between CD19[high] and CD19[low] compartments. a** Number of clonal families of BCRs found within the different BMPC clusters (diversity) and probability of finding different clonal families by random selection of cells (Simpson diversity index). A clonal family was defined by V and J gene composition and a CDR3 region with <20% Hamming distance in both the heavy and light chains originating from one donor. **b** Mutation rates in the framework regions FR1-3 (left) and in the CDR1-3 regions (right) of the heavy and light chain rearrangements across BMPC clusters. **c** Bubble plot of the mutation rates in the framework regions FR1-3 (left) and in the CDR1-3 regions (right) of BCRs per isotype and cluster. Colour scale indicates median mutation rates. Values below or above the scale limits are shown in blue or red, respectively. Bubble sizes correspond to the percentage of cells expressing a defined isotype within the indicated cluster. **d** Violin plot depicting the mutation rates in the V gene of the BCRs of BMPC per cluster. Statistical significance between clusters is shown in Supplementary Data 1 (two-tailed Mann–Whitney *U* test). Horizontal lines represent the median mutation rate. Violins are coloured by the *z* score of CD19 gene expression in each cluster. **e** Identification of SARS-CoV-2 spike-specific and tetanus toxoid-specific public clones (in black) among analysed BMPC. Public clones were defined by exhibiting over 80% CDR3 sequence identity in both heavy and light chains when compared to the BCR of sequenced peripheral blood and bone marrow spike- and tetanus-specific cells from vaccinated individuals (see Supplementary Fig. 1a and Supplementary Data 1). **f** Relative distribution of spike-specific (red) and tetanus-specific (blue) BMPC (depicted in **e**) per cluster. **g** Comparison of mutation rates within the framework regions FR1-3 (left) and in the CDR1-3 regions (right) of spike-specific (red) and tetanus-specific (blue) BMPC per isotype. Horizontal lines represent the median mutation rate. Statistics were performed using a two-tailed Mann–Whitney *U* test. Source data are provided as a Source Data file.

BNT and primary vaccination with AZ. Later on, most PC are recruited to CD19[high] clans.

## Vaccination-induced ASC reflect cytokine instructions

To identify the signals responsible for the differential transcriptional signatures of ASC and BMPC generated by vaccination, we compared these signatures to transcriptomes of B cells activated ex vivo under different conditions. The B cells were stimulated with anti-CD40 (to mimic T-cell help) and various combinations of cytokines for different durations[34,35]. Obtained gene signatures were projected by GSEA onto the UMAP of peripheral ASC collected at different times post vaccination (Fig. 4a) and onto the UMAP of BMPC (Fig. 4b, Supplementary Fig. 6a). While IL-6 and IL-21 gene signatures at 12 and 24 h highlight the proliferating peripheral ASC (Fig. 4a), they were mainly enriched in clans 5 and 13 of BMPC (Fig. 4b, Supplementary Fig. 6a). Signatures of B cells stimulated with IFN-α for 12 h marked BMPC of clan 8 (see also Fig. 1d, g), whereas interestingly, 24 h of IFN-α stimulation resulted in the marking of BMPC clans 4 and 6 (Fig. 4b, Supplementary Fig. 6a). B cells activated in the presence of IFN-α, TGF-β, IL-6 and IL-21 for 12 h showed signatures comparable to BMPC of clans 1 and 4 and mainly the HLA-DMA[high] cells of the peripheral ASC (Figs. 4a, b and 3b). After 24 h the signatures resembled those of clan 0 (Fig. 4b). Accordingly, the transcriptomes of ASC generated at defined time points after primary, secondary and tertiary vaccination do differ (Supplementary Fig. 6b), indicating the dynamic evolution of the immune response, i.e. the cytokines involved, and confirming the assignment of BMPC clans to its distinct phases.

## IL-21 downregulates expression of CD19 on activated B cells

Notably, B cells activated ex vivo with signals mimicking T-cell help, i.e. CD40L and IL-21, developed transcriptional signatures similar to BMPC from clans 1 and 4, i.e. CD19[low] BMPC. To test the hypothesis that CD19[low] ASC and BMPC are indeed generated in T cell-dependent follicular immune reactions, we sorted either tonsillar follicular T helper cells (CD45RA[−]CXCR5[high]) or peripheral T helper cells (CD45RA[−]CXCR5[−]) from bronchoalveolar lavage (BAL) and cultured them with tonsillar memory B cells (IgD[−]CD38[−]) in the presence of staphylococcal enterotoxin B (SEB)[36]. To block CD40/CD40L interactions and IL-21 receptor-induced signalling of the activated B cells, anti-CD40L and/or soluble IL-21 receptor (sIL-21R) were also added to the cultures. After 7 days of culture, we analysed the expression of CD19 on the CD27/CD38-expressing ASC generated (Fig. 4c, d, Supplementary Fig. 7a, b). Whether or not CD40-CD40L interaction was blocked, the resulting ASC were CD19[low]. In contrast, blocking IL-21 signalling inhibited CD19 downregulation in the ASC, with a 2- to 15-fold increase in the frequency of CD19[high] cells, for the 4 donors analysed (Fig. 4c, d). The IL-21-mediated downregulation of CD19 expression on activated B cells implies that CD19[low] BMPC are derived from follicular immune reactions, while CD19[high] BMPC are generated in extrafollicular immune reactions in the absence of IL-21 expressing follicular T helper

cells. This is in line with the fact that the CD19[low] compartment is enriched in cells exhibiting hallmarks of STAT3 signalling (Fig. 1g), known to be induced by IL-21 in germinal centres[37,38].

## Sequential recruitment of BMPC to the CD19[low] and CD19[high] compartments in the tertiary immune reaction to Comirnaty

We also analysed SARS-CoV-2 spike-specific and TT-specific BMPC by multiparametric flow cytometry. For a total of 20 BM samples (Supplementary Table 1) we identified both RBD of spike- and TT-specific plasma cells by intracellular staining with the respective antigen, using two fluorescent conjugates each, to ensure specificity of the staining (Fig. 5a, Supplementary Fig. 8a). The frequencies of RBD- and TT-specific BMPC did not significantly differ, with medians of 0.092% and 0.16%, respectively (Fig. 5b). These numbers are consistent with previous observations[19,39]. However, while the majority of RBD-specific plasma cells (67%) were CD19[high], the majority of TT-specific plasma cells (63%) were CD19[low] (Fig. 5c and Supplementary Fig. 8b). 75% of both RBD- and TT-specific BMPC expressed IgG, while 22% of RBD- and only 5% of TT-specific cells expressed IgA (Fig. 5d, Supplementary Fig. 8c). The remaining cells expressed IgM. Interestingly, with time elapsed after the third COVID-19 vaccination, frequencies of CD19[low] RBD-specific did increase, from about 20% one month after the third vaccination to more than 40% two months later (Fig. 5e). This increase is in line with the notion that secondary (and tertiary) immune reactions start with the extrafollicular reactivation of memory B cells, followed by generation of new and adapted ASC in a subsequent follicular immune reaction[40].

## Discussion

Here we provide a comprehensive single-cell gene expression analysis of human bone marrow plasma cells (BMPC). We find a remarkable similarity between 8 individuals, each with 10 clans of BMPC, characterised by transcriptomic signatures reflecting their imprinting in different types of immune reactions. We track the recruitment of PC precursors to the BM by comparing transcriptional signatures of ASC, newly generated in immune reactions to defined vaccines against SARS-CoV-2 spike protein or diphtheria/tetanus/pertussis, to transcriptional signatures of BMPC and human B cells activated ex vivo with defined stimuli.

ASC released from secondary immune reactions are detected in the blood, peaking after about 7 days, while those of primary immune reactions are detected a few days later[19,41,42]. Signatures of ASC released from primary and secondary immune reactions differ significantly and indicate sequential recruitment of BMPC to different clans. In primary and secondary immune reactions, which were induced closely following the primary response, BMPC are recruited to CD19[low] clans, while in true secondary and tertiary reactions, BMPC are recruited nearly exclusively to CD19[high] clans. We show that CD19 is downregulated by IL-21, an essential signal of follicular helper T cells in follicular immune reactions. Thus, while in the initial immune reaction

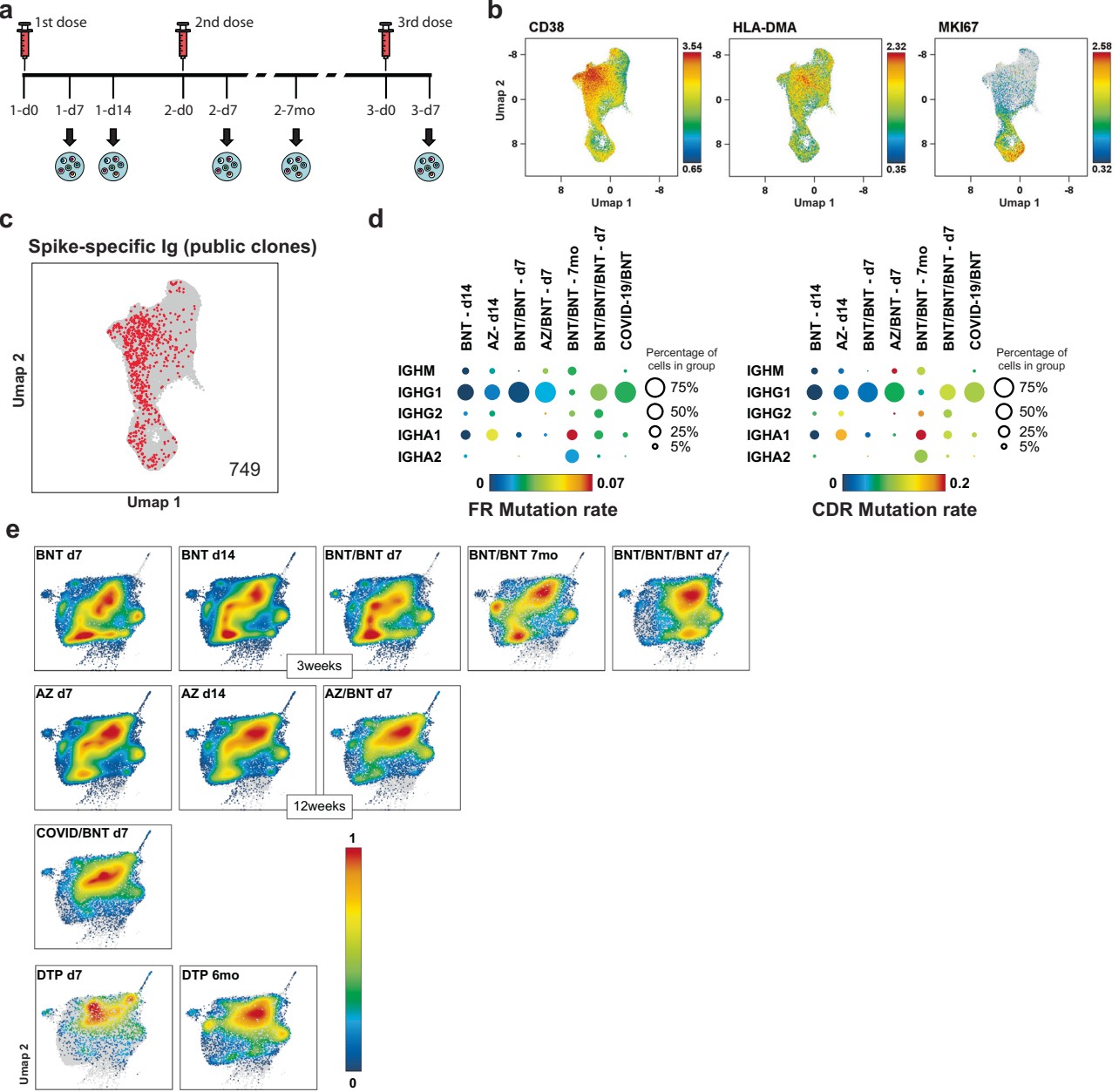

**Fig. 3 | Type and timing of B-cell activation imprints on BMPC subsets.**
**a** Schematic representation of vaccination, blood collection and sample analysis. Arrows indicate time points when transcriptome and full-length B-cell receptor repertoire sequencing was performed. **b** UMAP representation of the expression levels of selected genes in 55071 CD27$^{high}$CD38$^{high}$ sorted peripheral blood ASC from 36 healthy individuals after COVID-19 or diphtheria, tetanus, pertussis (DTP) vaccination (see Supplementary Table 2 for information on participants and different vaccination schemes). Colour scale represents the expression level of the indicated genes. **c** Identification of SARS-CoV-2 spike-specific (red) public clones among analysed ASC. Public clones were defined by displaying >80% CDR3 sequence identity when compared to the BCR of sequenced peripheral blood and bone marrow spike- and tetanus-specific cells from vaccinated individuals (see Supplementary Fig. 1a and Supplementary Data 1). The number of public clones found is shown on the UMAP. **d** Bubble plot of mutation rates in the framework regions (FR1-3, left) and in the CDR regions (CDR1-3, right) of identified spike-specific public clones per isotype identified in time point after COVID-19 vaccination. Colour scale indicates median mutation rates. Values above the scale limits are shown in red. Bubble sizes correspond to the percentage of spike-specific cells expressing a defined isotype within the indicated group. **e** Density plots of BMPC significantly enriched in gene signatures from peripheral blood ASC isolated at different time points after vaccination as identified by Gene Set Enrichment Analysis (GSEA). Violin plots of the normalised enrichment score (NES) per BMPC cluster are depicted in Supplementary Fig. 4a. Statistical significance between NES scores is shown in Supplementary Data 1 (two-tailed Mann–Whitney $U$ test). Source data are provided as a Source Data file.

BMPC are mostly derived from follicular immune responses, in subsequent immune reactions, they are mostly derived from extra-follicular immune responses. The incremental increase in CD19$^{low}$ spike-specific BMPC over time after the third vaccination with BNT suggests that after the extrafollicular reactivation of memory B cells, there is a continued follicular immune reaction that follows.

Additionally, the transcriptional signatures reveal instruction by interferon and later also by TGF-β in response to vaccination. In the end, the recruitment of BMPC in vaccinations to SARS-CoV-2 spike or tetanus toxoid generates BMPC of most clans in all individuals, reflecting the evolution of a given humoral immune response to sequential challenges. This ensures a broad immunity in terms of

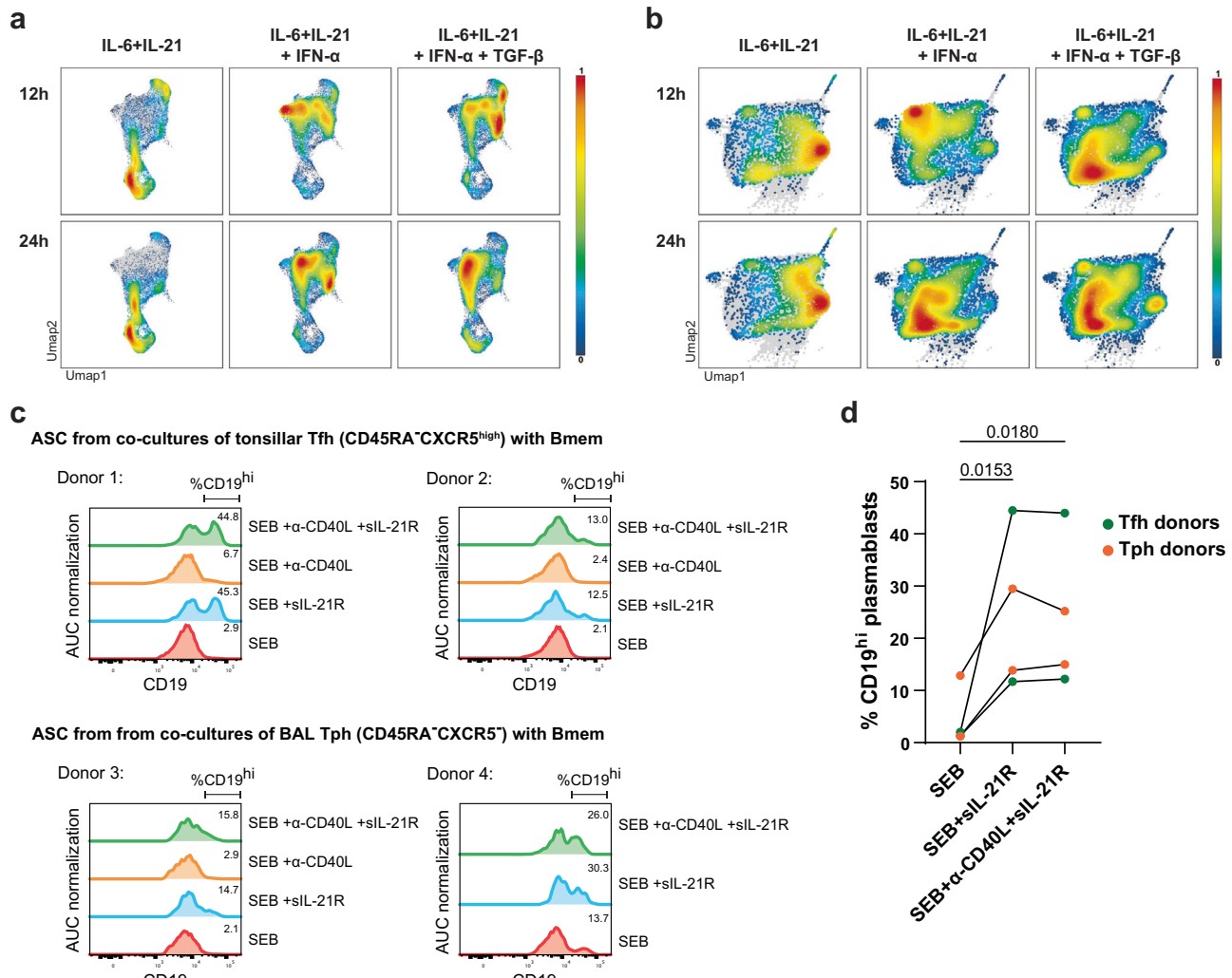

**Fig. 4 | Follicular immune responses involving IL-21 leads to downregulation of CD19 in ASC. a**, **b** Gene Set Enrichment Analysis (GSEA) based on gene signatures from ex vivo-differentiated plasmablasts in presence of different cytokine combinations (data sets from Stephenson et al.[35]). Briefly, naive B cells were cultured in differentiating conditions for 6 days after which different cytokines were added to the culture. After 12 or 24 h, cells were harvested and sequenced. Gene signatures were identified by comparing the different conditions and time points to 0 h. **a** Density plots of ASC (time points and vaccine protocols combined) with significant enrichment identified by GSEA of gene signatures from ex vivo-differentiated plasmablasts. **b** Density plots of BMPC with significant enrichment identified by GSEA of gene signatures from ex vivo-differentiated plasmablasts. Violin plots of the normalised enrichment score (NES) per BMPC cluster are depicted in Supplementary Fig. 6a. Statistical significance between NES scores is shown in Supplementary Data 1 (two-tailed Mann–Whitney $U$ test). **c**, **d** T follicular helper cells (Tfh, CD19$^-$CD4$^+$CD45RA$^-$CXCR5$^{++}$) from tonsils or T peripheral helper cells (Tph, CD19$^-$CD4$^+$CD45RA$^-$CXCR5$^-$) from BAL of sarcoidosis patients were co-cultured 1:1 with tonsillar memory B cells (CD19$^+$CD4$^-$IgD$^-$CD38$^-$) for 7 days in presence of staphylococcal enterotoxin B (SEB). Where indicated, T-cell help was blocked with an anti-CD40L antibody and/or recombinant soluble IL-21R. **c** Histograms of CD19 expression by differentiated plasmablasts (see Supplementary Fig. 7). Two Tfh and two Tph donors were analysed. **d** Frequencies of CD19$^{hi}$ plasmablasts. Statistics were performed using an RM parametric one-way ANOVA followed by multiple testing according to Fischer's LSD test. Source data are provided as a Source Data file.

antibody function, output of mature cells from follicular reactions, and a fast extrafollicular recall response.

The SARS-CoV-2 pandemic has revealed a remarkable illiteracy on immunological memory, the molecular and cellular basis of adaptive immunity. The persistence of humoral immunity, i.e. specific serum antibodies, has been demonstrated[3,6] as well as the decades-long persistence of the cells secreting them: the plasma cells[43,44]. The bone marrow is known to maintain long-lived plasma cells[2,3]. Bone marrow-resident plasma cells secreting spike protein-specific antibodies have been demonstrated in SARS-CoV-2 convalescent and vaccinated persons[4,5,45] at frequencies about equal to those secreting tetanus/ diphtheria-specific antibodies. Although it is already evident that BMPC are derived from different types of immune reactions from the class of the antibodies they secrete, little is known about their heterogeneity. With the advent of single-cell transcriptomics, different

studies on the transcriptional heterogeneity of murine and human BMPC have been published[46,47]. Here we show that distinct immune reactions, exemplified for SARS-CoV-2 spike and tetanus, are diverse and dynamic, and that specific BMPC are found in most vaccinees in several, if not all, compartments of BMPC. Moreover, in each subsequent challenge with the vaccine, PC are delegated to different compartments. This not only reflects the heterogeneity of instruction in each immune reaction, but also a sequential dominance of extrafollicular over follicular immune reactions.

Of particular interest is our observation that in primary immune reactions, and the closely-followed secondary vaccination with BNT, ASC exiting the immune reaction 7 days after vaccination show a transcriptional gene expression signature resembling that of clans 1 and 4 BMPC, including downregulated CD19 expression. After the original description of CD19$^{low}$ and CD19$^{high}$ BMPC[7], came studies

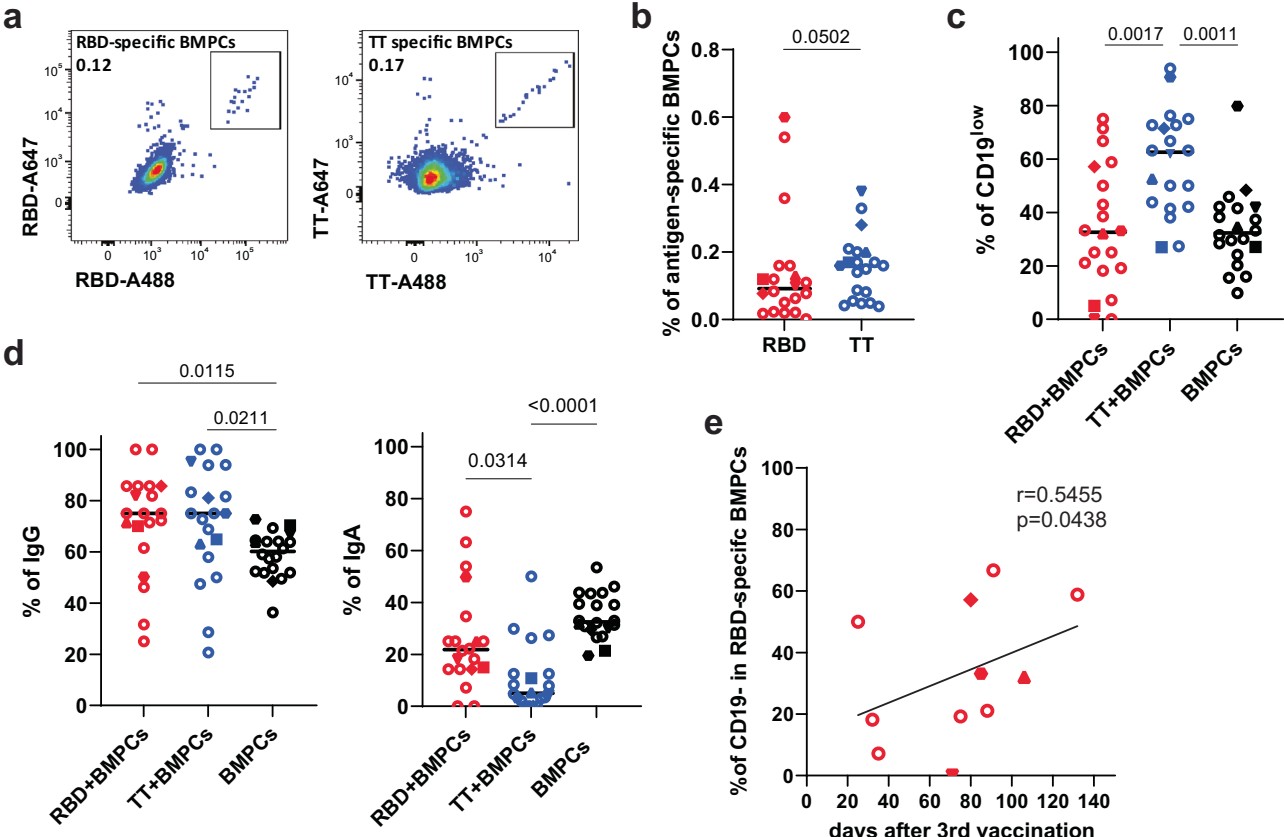

**Fig. 5 | Antigen-specific BMPC are sequentially recruited to both CD19$^{high}$ and CD19$^{low}$ BMPC compartments. a** Representative pseudocolour plots of intracellular double-positive SARS-CoV-2 RBD (left) or tetanus toxoid (TT, right) staining in CD38$^{high}$CD138$^{+}$CD14$^{-}$CD3$^{-}$ live singlet BMPC (gating strategy in Supplementary Fig. 8a). **b**–**e** Each symbol represents one donor/sample (see Supplementary Table 1). Filled symbols represent BMPC samples which were also analysed by single-cell sequencing (Fig. 1a). **b** Frequencies of RBD-specific and TT-specific BMPC within total BMPC. Horizontal lines indicate the median. Statistics were performed using the one-tailed Mann–Whitney $U$ test. $n = 20$ BM samples. **c** Frequencies of CD19$^{low}$ cells within RBD-specific BMPC (red), TT-specific BMPC (blue) and total

BMPC (black). Horizontal lines indicate the median. Statistics were performed using the Kruskal–Wallis tests with Dunn's correction for multiple comparisons. $n = 20$ BM samples. **d** Frequencies of IgG+ (left) and IgA+ (right) cells within RBD-specific BMPC (red), TT-specific BMPC (blue) and total BMPC (black). Horizontal lines indicate the median. Statistics were performed using the Kruskal–Wallis tests with Dunn's correction for multiple comparisons. $n = 20$ BM samples. **e** Correlation between the frequency of CD19$^{low}$ RBD-specific BMPC and days after 3rd vaccination against SARS-CoV-2. Statistics were performed using one-tailed Spearman's correlations. $n = 11$ BM samples. Source data are provided as a Source Data file.

showing that CD19$^{low}$ BMPC are refractory to B-cell depletion by rituximab[28], secrete antibodies against unique childhood experiences like measles and mumps, and are resistant to CD19-specific CAR T cells[48], defining CD19$^{low}$ BMPC as long-lived[49]. The role of CD19$^{high}$ BMPC was, however, less clear. Here we demonstrate that ASC downregulate CD19 upon instruction by IL-21 in follicular immune reactions. This defines CD19$^{high}$ BMPC clusters as being of extrafollicular origin, while CD19$^{low}$ BMPC are of follicular origin. CD19 has been implied in supporting ERK signalling and AKT phosphorylation in human plasma cells[50] and also CXCR4 signalling in IgD-deficient B cells[51]. We have previously described, that CXCR4 can support the survival of BMPC ex vivo[52] and in vivo[53]. Moreover, we have demonstrated the pivotal role of the PI3K/AKT/Foxo1/3a signalling axis for survival of BMPC, preventing activation of caspases 3 and 7 by metabolic stress[23]. On the level of single cell transcriptomes, both CD19$^{low}$ and CD19$^{high}$ BMPC show genes associated with resilience to apoptosis, namely, *MCL1*[54]. CD19$^{high}$ BMPC also thus qualify for being long-lived, but they in all likelihood originate from repeated antigenic challenge, such as influenza, as already demonstrated by Halliley and colleagues[49]. Using a systemic and innovative approach we demonstrate the recruitment of ASC generated upon primary and secondary vaccination with SARS-CoV-2 vaccines to CD19$^{low}$ and CD19$^{high}$ BMPC clans. In primary immune reactions to BNT, and the closely-followed boost, and to AZ, most ASC

are recruited to CD19$^{low}$ clans 1 and 4, while in true secondary immune reactions, most ASC generated at early time points are recruited to CD19$^{high}$ clans, in particular clan 0. At later time points, ASC may also be recruited from follicular immune reactions, as is indicated by an increase in antigen-specific CD19$^{low}$ BMPC over a timespan of several months.

IL-21 is a signature cytokine of follicular T helper cells, which orchestrate B cell activation in germinal centre reactions[55–58]. It impacts proliferation, affinity maturation and memory B-cell generation in a paracrine, synapse-independent way, and at high concentrations, also affects the differentiation of activated B cells into plasmablasts[38,57,59]. This regulation is B cell intrinsic[56], although the precise gene regulatory mechanisms are not fully understood. As we show here, IL-21 also (down)regulates expression of CD19 on B cells activated to differentiate into ASC. Accordingly, we observe recruitment of ASC generated in primary immune responses induced by SARS-CoV-2 vaccines to CD19$^{low}$ BMPC clans. Also, IL-21's influence in germinal centre plasma cell differentiation has been shown to occur via a robust and prolonged STAT3 activation[37,38], which is consistent with the respective GSEA highlighting BMPC belonging to the CD19$^{low}$ compartment. In primary responses, plasma cells are expected to be generated from naïve B cells activated in germinal centres, whereas in secondary immune reactions, extrafollicular reactivation of memory B

cells generated in previous immune reactions has been postulated as a way of delivering an immediate burst of specific antibody production, as imprinted in the previous immune reaction[60]. While reports of both IL-21-dependent and -independent extrafollicular reactions have been described[61–63], in the immune responses here analysed extrafollicular reactivation was IL-21-independent, since the ASC generated maintained CD19 expression and their canonical transcriptional signature allocates to CD19[high] BMPC. Indeed, in the donors analysed, CD19[high] BMPC made up as much as 20.2 to 90.1% of all BMPC. There is no evidence that these BMPC are not as long-lived as CD19[low] BMPC, and do not use the same molecular mechanisms to persist[23]. CD19 could even be expected to support the antigen-receptor-independent, stromal cell contact-dependent survival of these cells. Indeed, we found substantial populations of tetanus-specific CD19[high] BMPC in the bone marrow of the donors analysed.

The derivation of BMPC from follicular and extrafollicular immune reactions has fundamental implications for understanding imprinting and regulation of (humoral) immunological memory. Feedback-regulation by pre-existing specific antibodies was first demonstrated by T. Smith in 1909[64], and more recently by the group of Nussenzweig[65], who showed that vaccinees treated with SARS-CoV-2-specific therapeutic antibodies before, upon vaccination, did not respond to the epitopes recognised by these antibodies. The rapid extrafollicular reactivation of memory B cells generated in previous immune reactions thus shapes secondary immune responses in providing immediate, imprinted immunity to the original antigen (original antigenic sin)[66] and blocking reactions to epitopes recognised by them[67,68]. As we show here, ASC of these imprinted reactions are also recruited to the BM, providing long-lasting enhanced immunity. Adaptation to variants of the original antigen may occur later, as is suggested here by the increasing frequency of CD19[low] SARS-CoV-2-specific BMPC months after the tertiary vaccination with BNT. It remains to be shown to what extent this tertiary germinal centre reaction is regulated by the antibodies produced in the fast extrafollicular reaction, but also, to what extent the extrafollicular reaction is regulated by the pre-existing antibodies of already established BMPC. Understanding the interplay of follicular and extrafollicular immune reactions in shaping the repertoire of BMPC will be key to develop vaccines and vaccination protocols for the establishment of sustainable and lasting immunity, and the adaptation of immunity to newly arising variants.

## Methods
### Human donors
The recruitment of study participants was conducted in accordance with the Ethics Committee of the Charité Universitätsmedizin Berlin in compliance with the Declaration of Helsinki (EA1/261/09, EA1/009/17 and EA2/010/21). Informed consent was obtained from all bone marrow/peripheral blood donors included in the study. No compensation was provided to participants. Bone marrow samples were obtained from 25 patients undergoing total hip arthroplasty without any underlying malignant or inflammatory disease (12 females and 13 males with a median age of 62 years, see Supplementary Table 1). Peripheral blood was obtained from 36 healthy volunteers at different time points after Comirnaty, Vaxzevria or Boostrix vaccination (16 females and 19 males with a median age of 32 years, see Supplementary Table 2). Due to the small number of analysed samples, no disaggregated sex or gender analysis was performed.

### Bone marrow plasma cell (BMPC) isolation
Bone marrow samples were fragmented and transferred to a 50 mL tube where they were vortexed to separate cells from bone fragments. Samples were subsequently rinsed through a 70 μM filter with PBS/1% BSA/5 mM EDTA/2 μg/mL actinomycin D to obtain a cell suspension. Plasma cells were enriched from bone marrow using StraightFrom

Whole Blood and Bone Marrow CD138 MicroBeads and StraightFrom Whole Blood CD19 MicroBeads (Miltenyi Biotec) according to manufacturer's instructions. Enriched cells were incubated with Fc Blocking Reagent (Miltenyi Biotec) following manufacturer's instructions and subsequently stained for 30 min at 4 °C with the following anti-human antibodies: CD3-VioBlue (BW264/56, Miltenyi Biotec, Cat. 130-113-133, 1:400), CD10-VioBlue (97C5, Miltenyi Biotec, Cat. 130-099-670, 1:11); CD14-VioBlue (TÜK4, Miltenyi Biotec, Cat. 130-113-152, 1:200); CD38-APC (HIT2, BioLegend, Cat. 303510, 1:25) and CD138-PE (44F9, Miltenyi Biotec, Cat. 130-119-840, 1:50) or CD3-VioBlue (BW264/56, Miltenyi Biotec, Cat. 130-113-133, 1:400), CD14-VioBlue (TÜK4, Miltenyi Biotec, Cat. 130-113-133, 1:200), CD27-APC-Cy7 (O323, BioLegend, Cat. 302816, 1:25), CD38-PerCP-Cy5.5 (HIT2, BD Biosciences, Cat. 551400, 1:100) and tetanus toxoid (AJ vaccines) coupled with Alexa Fluor 647 or Alexa Fluor 488 and SARS-Cov2 Spike Protein (Biotin, Miltenyi Biotec, Cat. 130-127-682) pre-incubated with streptavidin PE or streptavidin PE-Cy7 according to manufacturer's instructions. Simultaneously, cells were incubated with DNA barcoded antibodies for Cellular Indexing of Transcriptomes and Epitopes by Sequencing (CITE-seq, see antibody list). DAPI was added before sorting to allow dead cell exclusion. Two different gating strategies were used for sorting (CD138[+]CD38[high] or CD27[high]CD38[high], see Supplementary Fig. 1a), depending on the need to sort either bulk plasma cells or also antigen-specific memory B cells. Memory B cells were also sorted but not included in this analysis. All sortings were performed using a MA900 Multi-Application Cell Sorter (Sony Biotechnology). Cell counting was performed using a MACSQuant flow cytometer (Miltenyi Biotec). The sorted cells were further processed for single-cell RNA sequencing.

### Peripheral blood antibody-secreting cell (ASC) isolation
Cells were enriched by positive selection from peripheral blood using a combination of StraightFrom Whole Blood CD19 and CD3 MicroBeads and StraightFrom Whole Blood and BM CD138 MicroBeads (Miltenyi Biotec) according to manufacturer's instructions. 2 μg/mL actinomycin D was added to the buffer used during the first centrifugation. Enriched cells were incubated with Fc Blocking Reagent (Miltenyi Biotec) following manufacturer's instructions and subsequently stained for 30 min at 4 °C with the following anti-human antibodies: CD3-FITC (UCHT1, DRFZ in-house, 1:10), CD14-VioBlue (TÜK4, Miltenyi Biotec, Cat. 130-113-152, 1:200), CD27-PE (MT271, Miltenyi Biotec, Cat. 130-113-630, 1:100) and CD38-APC (HIT2, BioLegend, Cat. 303510, 1:25). Simultaneously, cells were incubated with DNA barcoded antibodies for Cellular Indexing of Transcriptomes and Epitopes by Sequencing (CITE-seq), which allowed identification of samples from different donors (see antibody list, hashtags). DAPI was added before sorting to allow dead cell exclusion. See Supplementary Fig. 3a for gating strategy. While B cells and T cells were also isolated, they were not included into the analysis. All sortings were performed using a MA900 Multi-Application Cell Sorter (Sony Biotechnology). Cell counting was performed using a MACSQuant flow cytometer (Miltenyi Biotec). The sorted cells were further processed for single-cell RNA sequencing.

### Single-cell RNA-library preparation and sequencing
Single-cell suspensions were obtained by cell sorting and applied to the 10x Genomics workflow for cell capturing and scRNA gene expression (GEX), BCR and CITE-Seq library preparation using the Chromium Single Cell 5′ Library & Gel Bead Kit version 2 for BMPC or version 1.1 for ASC, as well as the Single Cell 5′ Feature Barcode Library Kit (10x Genomics). After cDNA amplification, the CITE-Seq libraries were prepared separately using the Dual Index Kit TN Set A for BMPC or the Single Index Kit N Set A for ASC. BCR target enrichment was performed using the Chromium Single Cell V(D)J Enrichment Kit for Human B cells. Final GEX and BCR libraries were obtained after fragmentation, adapter ligation and final Index PCR using the Dual Index

Kit TT Set A for BMPC or the Single Index Kit T Set A for ASC. Qubit HS DNA assay kit (Life Technologies) was used for library quantification and fragment sizes were determined using the Fragment Analyzer with the HS NGS Fragment Kit (1-6000 bp) (Agilent).

Sequencing was performed on a NextSeq2000 device (Illumina) applying the sequencing conditions recommended by 10x Genomics for libraries prepared with Next GEM Reagent Kits. NEXTSeq 1000/2000 P3 reagent kits (200 Cycles, Illumina) were used for 5′ GEX and CITE-Seq libraries (for version 2, read1: 26nt, read2: 90nt, index1: 10nt, index2: 10nt; for version 1.1, read1: 26nt, read2: 98nt, index1: 8nt, index2: 0nt) and NEXTSeq 1000/2000 P3 reagent kits (300 Cycles, Illumina) were used for BCR libraries (for version 2, read1: 151nt, read2: 151nt, index1: 10nt, index2: 10nt., 2% PhiX spike-in, for version 1.1, read1: 151nt, read2: 151nt, index1: 8nt, index2: 0nt., 2% PhiX spike-in).

### Single-cell transcriptome analysis

Raw sequence reads were processed using cellranger (version 5.0.0). Demultiplexing, mapping, detection of intact cells as well as quantification of gene expression was performed using cellranger's count pipeline in default parameter settings with refdata-cellranger-hg19-1.2.0 as reference and expected number of 3000 cells per sample. Noteworthy, the used reference does not contain immunoglobulin genes as defined by the respective biotype, which would otherwise strongly influence the downstream analysis. An average of 48000 mean reads per cell was obtained, which is more than double of the 20000 reads per cell recommended by the manufacturer for 10x genomics 5′ version 2. (https://kb.10xgenomics.com/hc/en-us/articles/115002022743-What-is-the-recommended-sequencing-depth-for-Single-Cell-3-and-5-Gene-Expression-libraries-). This led to 20378, 15322, 10818, 10707, 8876, 8394, 9209 and 6206 intact cells for 8 bone marrow samples. Cellranger's aggr was used to merge the libraries without size normalisation and to perform a Uniform Manifold Approximation and Projection (UMAP). Loupe Browser (version 5, 10x Genomics) was used to identify and define bone marrow plasma cells (BMPC) by manual gating. Plasma cells defined clear regions with cells expressing *PRDM1*, *SDC1*, *XBP1* and *IRF4* genes. This led to 11052, 10772, 5505, 5661, 2222, 5528, 4556, 4051 plasma cells from each of the 8 BM samples. The BMPC were further analysed in R (version 4.1.2) using the Seurat package (version 4.0.5)[69] and the cellranger's aggr output and the respective cellular barcodes. In particular, the transcriptome profiles of the BM samples were read and plasma cells were extracted using Read10x, CreateSeuratObject and subset. To identify cells with similar transcriptional profiles among different sequencing libraries, sample specific batch effects were removed as described in FindIntegrationAnchors (Seurat) R Documentation. In particular, samples were analysed individually using SplitObject by LibraryID, NormalizeData with LogNormalization as normalisation.method and scale.factor of 10,000, FindVariableFeatures with 2000 variable genes and vst as selection method, ScaleData and finally RunPCA to compute 50 principle components for each sample. Next, common anchors were identified by FindIntegrationAnchors usind rpca as reduction, 2000 anchor.features and 1:30 dimensions and finally merged using IntegrateData. Based on the integrated data, a uniform manifold approximation and projection (UMAP) was computed using ScaleData, RunPCA to compute 50 principle components and RunUMAP using 1:30 dimensions. Transcriptionally similar clusters were identified by shared nearest neighbour (SNN) modularity optimisation using FindNeighbors with pca as reduction and 1:30 dimensions as well as FindClusters with resolutions ranging from 0.1 to 1.0 in 0.1 increments using FindCluster. Further analyses were based on non-integrated, log normalised values represented as ln (10,000 × UMIsGene)/UMIsTotal +1) and the above integrations-based clusters and UMAP. By visual inspection of the percentage of mitochondrial genes, UMI counts, number of identified genes as well as expression of typical marker genes projected on the UMAP, clustering with a resolution of 0.5 was judged to best reflect the transcriptional community structure. Clusters comprising low

quality cells as well as clusters comprising contaminations were not considered in further analyses. For the pseudotype trajectory analysis the considered BMPC clusters were reintegrated as described above. Pseudotime trajectory analysis was performed by applying the default pipeline of the R package monocle3[70], with the exception of using the integrated data as input matrix[71].

The single-cell transcriptome analysis of blood ASC was performed in accordance to the BMPC analysis. In particular, libraries of pooled samples from 36 individuals at different time points (see Supplementary Table 2) were demultiplexed and mapped, intact cells were detected, and gene expression was quantified by cellranger's count pipeline and merged by cellranger aggr. This led to 32162 and 32506 intact cells at d7 and d14 after Comirnaty 1st dose of (BNT d7, d14), respectively; 42047 and 26339 cells at d7 and 7 months after Comirnaty 2nd dose (BNT/BNT d7, 7mo), respectively; 32860 cells at d7 after Comirnaty 3rd dose (BNT/BNT/BNT d7); 28282 and 24622 cells at d7 and d14 after Vaxzevria 1st dose (AZ d7, d14), respectively; 16567 cells at d7 after Comirnaty 2nd dose (1st dose Vaxzevria, AZ/BNT d7); 21524 cells at d7 after 1st Comirnaty dose of donors recovered from a SARS-CoV-2 infection (COVID/BNT d7); and 29875 and 22811 cells at d7 and 6mo after Boostrix boost (DTP d7, 6mo), respectively. Libraries were merged by cell ranger's aggr. Loupe Browser (version 5, 10x Genomics) was used to identify and define ASC by manual gating. ASC defined clear regions with cells expressing higher levels of *PRDM1*, *CD27* and *CD38* genes. This led to 4462 and 5762 ASC at d7 and d14 after Comirnaty 1st dose of (BNT d7, d14), respectively; 5983 and 4169 ASC at d7 and 7 months after Comirnaty 2nd dose (BNT/BNT d7, 7mo), respectively; 6036 ASC at d7 after Comirnaty 3rd dose (BNT/BNT/BNT d7); 8540 and 2143 ASC at d7 and d14 after Vaxzevria 1st dose (AZ d7, d14), respectively; 2969 ASC at d7 after Comirnaty 2nd dose (1st dose Vaxzevria, AZ/BNT d7); 5586 ASC at d7 after 1st Comirnaty dose of donors recovered from a SARS-CoV-2 infection (COVID/BNT d7); and 5679 and 3742 ASC at d7 and 6mo after Boostrix boost (DTP d7, 6mo), respectively. ASC were further analysed with the Seurat package R package using the cellranger's aggr output and the respective cellular barcodes. UMAP was performed by removal of library and donor specific batch effects using the Seurat's integration as described above for BMPC. Visualised is the log normalised gene expression of non-integrated data.

The Integration of the filtered BMPC clusters and blood ASC (Supplementary Fig. 3e) was performed as describe above, considering LibraryID and Donor.

### Single-cell immune profiling

Raw sequence reads were processed using cellranger (version 5.0.0). Vdj was used in default parameter settings for demultiplexing and assembly of the B cell receptor sequences using refdata-cellranger-vdj-GRCh38-alts-ensembl-2.0.0 as reference. The cellranger output was further analysed in R (version 4.1.2) using the Seurat package (version 4.0.5)[69].

B cell receptor isotypes and receptor sequences were assigned to the corresponding cells in the single-cell transcriptome analysis by identical cellular barcodes. In case of multiple contigs, the most abundant, productive and fully sequenced contig for the heavy and light BCR chain was used. This led to the annotation of 95% (3856), 90% (4954), 90% (5097), 94% (2081), 87% (9551), 85% (9105), 91% (5032) and 93% (4237) plasma cells for the 8 BM samples; 69% (3060) and 67% (3860) ASC at d7 and d14 after Comirnaty 1st dose of (BNT d7, d14), respectively; 65% (3893) and 66% (2762) ASC at d7 and 7 months after Comirnaty 2nd dose (BNT/BNT d7, 7mo), respectively; 70% (4248) ASC at d7 after Comirnaty 3rd dose (BNT/BNT/BNT d7); 53% (4525) and 64% (1364) ASC at d7 and d14 after Vaxzevria 1st dose (AZ d7, d14), respectively; 65% (1931) ASC at d7 after Comirnaty 2nd dose (1st dose Vaxzevria, AZ/BNT d7); 61% (3405) ASC at d7 after 1st Comirnaty dose of donors recovered from a SARS-CoV-2 infection (COVID/BNT d7);

and 69% (3938) ASC at d7 after Boostrix boost (DTP d7). The high-confidence contig sequences with an associated transcriptional profile were reanalysed using the HighV-QUEST at IMGT web portal for immunoglobulin (IMGT) to retrieve the germline sequence between of the FR1-CDR1-FR2-CDR2-FR3, the V-, J- and D-gene information as well as the nucleotide and amino acid CDR3 sequence. The HighV-QUEST output, in particular the IMGT-gapped-nt-sequences and V-REGION-mutation-and-AA-change-table were used to reverse engineer the gapped germline FR1-CDR1-FR2-CDR2-FR3 sequence. A clonal family of a BCR receptor was defined by the germline FR1-CDR1-FR2-CDR2-FR3 sequence, the used VJ-genes, the length of the CDR3 sequence and CDR3 identity greater than 80% (modelled optimum for CDR3 specificity[31]) between all members of a clonal family considering both the heavy and light chain. The clonal family annotation was used to compute the diversity, Simpson Diversity Index as well as the overlap table. Significance of an overlap was evaluated by 1000 permutation of the clonal family annotation of the cells (see Supplementary Data 1 for statistics). Mutation counts in framework regions (FR1, FR2, FR3) were taken from V-REGION-nt-mutation-statistics of the HighV-QUEST output. Mutation rates were defined as the sum of the estimated mutation counts in the heavy and light chain normalised to the length of the corresponding nt sequence length.

Spike- and tetanus-specific clones were derived from BCR-sequencing of spike- and tetanus-specific B cells isolated with either fluorophore-coupled RBD/Spike protein of SARS-CoV-2 or tetanus toxoid after vaccination of healthy individuals with Comirnaty or Boostrix vaccines. In particular, raw sequence reads from BCR-sequencing were processed using cellranger vdj pipeline as described above. Solely contig pairs for the heavy and light chain were considered. Unpaired contigs, that is contigs without a corresponding contig with the same cellular barcode, were removed from further analyses. In case of multiple contigs for the same cellular barcode, the most abundant, productive and fully sequenced contig for the heavy and light BCR chain were used. Putative spike- and tetanus-specific clones in BMPC and ASC samples were identified by comparing them with the experimentally validated spike- and tetanus-specific B cell clones. To meet the selection criteria, clones were required to exhibit consistent lengths in both heavy and light chains within the CDR3 region. Additionally, samples were defined based on a minimum CDR3 nucleotide identity threshold, calculated as the sum of identical nucleotides normalised to the total length of these regions, with a minimum requirement of 80% identity. This stringent approach ensured that only clones meeting both criteria were considered putative spike- or tetanus-specific clones for further analysis (see Supplementary Data 1).

## Gene set enrichment analysis (GSEA)

GSEA was performed for each cell based on the difference to the mean of log normalised expression values of all cells in the analysed set as pre-ranked list and 1000 randomisations (PMID: 16199517, PMID: 12808457). Significant up- or downregulation was defined by a FDR ≤ 0.50 and normalised $p$ value < 0.05 43. For visualisation, NES for significant cells were plotted. The GSEA was performed for indicated cells using hallmark gene sets (PMID: 26771021), REACTOME (PMID: 29145629), KEGG (PMID: 10592173), ex vivo-differentiated plasmablast gene signatures (defined as described before[34,35]), as well as ASC time-specific gene sets and spike-specific ASC time-specific gene sets from the ASC analysis as defined by marker genes for different time points after vaccination (Supplementary Data 1). Hallmark gene sets, REACTOME and KEGG were obtained from the MSigDB Collections (PMID: 26771021).

For the ASC signature gene set samples at day 7 and day 14 after first Comirnaty vaccination, day 7 and 7 months after second Comirnaty vaccination, day 7 after third Comirnaty vaccination, day 7 and day 14 after Vaxzevria vaccination, day 7 after second

Comirnaty vaccination (heterologous), day 7 after Comirnaty vaccination from donors recovered from a SARS-CoV-2 infection, as well as samples at day 7 and 6 months after vaccination with Boostrix, genes with an Area under the ROC Curve greater than 0.6 and an adjusted $p$ value ≤0.05 (Mann–Whitney $U$ Test) were defined as marker genes. For the spike-specific ASC signature gene set samples at day 14 after first Comirnaty vaccination, day 7 and 7 months after second Comirnaty vaccination, day 7 after third Comirnaty vaccination and day 7 after Comirnaty vaccination from donors recovered from a SARS-CoV-2 infection, genes with an Area under the ROC Curve greater than 0.6 and an adjusted $p$ value ≤0.05 (Mann–Whitney $U$ Test) were defined as marker genes. No significant signature genes were found for spike-specific ASC obtained at day 14 after first Comirnaty vaccination.

The GSEA results were visualised by density plot on UMAPs and when appropriate also on violin plots of the NES score of significant enriched cells with a positive NES score. Differences in positive NES scores were evaluated using the Mann–Whitney U Test (see Supplementary Data 1 for statistics).

## BM mononuclear cells flow cytometry analysis

BM mononuclear cells were enriched by density gradient centrifugation over Ficoll-Paque PLUS (GE Healthcare BioSciences)[28]. Briefly, samples were fragmented, rinsed with PBS/0.5%BSA/EDTA (PBE) (Miltenyi Biotech). The collected BM mononuclear cells were filtered with a 70 μm cell strainer (BD Biosciences), and then washed twice with PBE for staining.

All flow cytometry analyses were performed using a BD FACS Fortessa (BD Biosciences). To ensure comparable mean fluorescence intensities over time of the analyses, Cytometer Setup and Tracking beads (BD Biosciences) and Rainbow Calibration Particles (BD Biosciences) were used. For staining, LIVE/DEAD Fixable Blue Dead Cell Stain Kit (ThermoFisher Scientific) was used to exclude dead cells according to the manufacturer's protocol. BM cells were surface-stained for 30 min at 4 °C with the following anti-human antibodies: CD138-BUV737 (MI15, BD Biosciences, Cat. 564393, 1:20), CD14-BUV395 (M5E2, BD Biosciences, Cat. 740286, 1:50), CD3-BUV395 (UCHT1, BD Biosciences, Cat. 563546, 1:50), CD27-BV786 (L128, BD Biosciences, Cat. 563328, 1:50), CD19-BV711 (SJ25C1, BD Biosciences, Cat. 563038, 1:50), CD20-BV510 (2H7, BioLegend, Cat. 302340, 1:50), IgD-PE/Dazzle594 (IA6-2, BioLegend, Cat. 348240, 1:500), CD38-APC-Cy7 (HIT2, BioLegend, Cat. 303534, 1:500), HLA-DR- PE (Tü36, BD Biosciences, Cat. 555561, 1:10), CD56-BV421 (HCD56, BioLegend, Cat. 318328, 1:25) diluted in Brilliant Stain buffer (BD Horizon). Cells were washed twice with PBE, fixed for 20 min at 4 °C using Fixation/Permeabilization Solution Kit (BD Cytofix/Cytoperm Plus) and washed twice with perm/wash buffer. Cells were then stained intracellularly for 30 min 4 °C with recombinant purified RBD (DAGC149, Creative Diagnostics, New York, USA) and TT (AJ vaccines), which were coupled with either Alexa Fluor 647 or Alexa Fluor 488 to identify antigen-specific cells as described above and in ref. 72,73, and with anti-human antibodies to detect expressed isotypes: IgA-biotin (G20-359, BD Biosciences, Cat. 555884, 1:50), IgG-PE-Cy7 (G18-145, BD Biosciences, Cat. 561298, 1:500), IgM-BV421 (G20-127, BD Biosciences, Cat. 562618, 1:100). Double-positive cells were considered as antigen-specific cells (See Fig. 5a). Flow cytometric data were analysed by FlowJo software 10.7.1 (TreeStar).

One-tail spearman's correlation coefficients were estimated to assess the relationship between the frequencies of CD19⁻ frequency of antigen-specific BMPCs and the time since the third SARS-CoV-2 vaccination. Mann–Whitney $U$ test was used for comparison of two groups and Kruskal-Wallis with Dunn's post-test was used for multiple comparisons. All statistical analyses were conducted using Prism version 9 (GraphPad), and $P$ values of <0.05 were considered significant.

## Enzyme-linked immunosorbent assay for the detection of serum-specific antibody titres on patients undergoing hip replacement surgery

To determine the tetanus toxoid and SARS-CoV-2 RBD-specific antibody titres, 96-well plates were coated overnight with 0.5 μg/ml of either tetanus toxoid (AJ vaccines) or SARS-CoV-2 (2019-nCoV) Spike RBD-His recombinant protein (Sino biological, Cat. 40592-V08B-100). Coated plates were washed, blocked for 1 h with PBS 5% BSA and incubated overnight at 4 °C with serial dilutions of sera. Specific IgA antibodies were detected using anti-human IgA-Biotin (Southern Biotech, Cat. 2050-08) followed by streptavidin-HRP (Invitrogen, Cat. N100) and specific IgG antibodies were detected using anti-human IgG-HRP (Southern Biotech, Cat. 2040-05). Detection antibody incubation was performed at room temperature for 1 h. After washing 5 times with PBS-T, Tetramethylbenzidine (TMB) Substrate (Invitrogen, Cat. 88-7324-88) was added. The reaction was stopped by the addition of 2 N H2SO4 (Sigma-Aldrich: Cat. 84736). Optical densities were measured on Spectramax (Molecular devices). Optical densities were measured on Spectramax plus 384(Molecular devices). OD values were further plotted against respective sample dilutions, and areas under the curve (AUC) were quantified using Graphpad Prism 9.3.1.

## Flow cytometric assay for the detection of serum-specific antibody titres on patients undergoing hip replacement surgery

HEK293T cells (ATCC CRL-3216) were transfected with a plasmid expressing wild-type SARS-CoV-2 S protein. Next day, the proportion of transfected cells was determined by staining with anti-SARS-CoV-2 Spike Glycoprotein S1 antibody (clone: CR3022, Abcam, Cat. ab273073) for 30 min, wash cells once with PBS/0.2 % BSA and subsequent staining with goat anti-human IgG-Alexa647 (Southern Biotech, Cat. 2014-31). Further transfected cells were collected and incubated with sera for 30 min, washed twice with PBS/BSA and stained with goat anti-human IgG-Alexa647 (Southern Biotech, Cat. 2014-31) and anti-human IgA FITC (Sothern Biotech, Cat. 2052-02). Cells were washed with PBS/0.2 % BSA and either measured directly, dead cell exclusion by DAPI or stained for dead cells with Zombie Violet (Biolegend, Cat. 423113) in PBS for 5 min at room temperature and fixed in 4 % paraformaldehyde solution overnight at 4 °C. Samples were acquired on a FACSCanto (BD Biosciences) or a MACSQuant 16 (Miltenyi) and analysed using FlowJo v10 (Tree Star Inc.) analysis software. In the respective fluorescent channels, the geometric mean of fluorescent intensity (MFI) of Spike expressing and non-expressing cells was quantified, and $\Delta MFI = MFI\ (S+) - MFI\ (S-)$ for IgG and IgA was determined. ΔMFI values were further plotted against respective serum dilutions, and AUC were quantified using Graphpad Prism 9.3.1.

## Enzyme-linked immunosorbent assay for the detection of serum-specific antibody titres on vaccinated individuals

The amount of SARS-CoV-2 spike RBD-specific antibodies was quantified using an in-house ELISA described previously[74]. Briefly, purified RBD protein was used for coating at a concentration of 5 μg/ml and 50 μl per well in a 96-well microtitre plate (Costar 3590, Corning Incorporated, Kennebunk, USA). For the ChAd-Y25 titre, the same ELISA approach was used, and $5 \times 10^8$ viral particles/well of the Vaxzevria vaccine (AstraZeneca, Oxford) were used for coating. After overnight coating at 4 °C, 230 μl of 10% FCS in PBS per well was used for blocking. Blocking was performed for 1 h at RT. Plates were washed four times with PBS-T (PBS containing 0.05% Tween). 50 μl of in blocking buffer 1:100 diluted sera were incubated in the wells for 1.5 h. An HRP-linked anti-human IgG antibody (Cytiva, Cat. NA933-1ML, Dassel, Germany) at a dilution of 1:3000 was used as a secondary antibody and incubated for 1.5 h on the plates. After washing five times, plates were developed for 5 min with 100 μl TMB solution (eBioscience, San Diego, USA) and stopped with the same volume of 1 N sulphuric acid. Absorbance was measured directly at 450 m on an Infinite M1000 reader (Tecan Group, Männedorf, Switzerland).

## Co-culture of helper T cells with memory B cells

Tonsillar follicular helper T cells (Tfh, CD19−CD4+CD45RA−CXCR5high) from patients who underwent routine tonsillectomy or peripheral memory T helper cells from bronchoalveolar lavage (BAL) of sarcoidosis patients (mostly peripheral helper T cells, Tph, CD19−CD4+CD45RA−) were sorted on an ARIA II flow cytometry sorter (Becton Dickinson). Patient samples were obtained from the Unfallkrankenhaus Marzahn (tonsils) or the Charité Universitätsmedizin Berlin (BAL). Sorted T cells were co-cultured for 7 days with heterologous tonsillar memory B cells (CD19+CD4−lgD−CD38−) at a 1:1 ratio in the presence of 4 ng/ml staphylococcal enterotoxin B (Toxin Technology) as described previously[36]. To block T cell help, 20 μg/ml anti-CD40L antibody (clone TRAP1) and/or 10 μg/ml recombinant soluble IL-21 receptor (R&D Systems, Cat. 9249-R2) were added to the culture. Cells were acquired on an LSR II Fortessa flow cytometer (Becton Dickinson) and analysed using FlowJo version 10 software (Tree Star Inc.).

All antibodies utilised in the study are listed in Supplementary Table 3 for reference.

## Reporting summary

Further information on research design is available in the Nature Portfolio Reporting Summary linked to this article.

## Data availability

Next Generation Sequencing data sets generated in this study are available in the Gene Expression Omnibus (GEO) repository under accession number GSE253862. Data was mapped using the human genome reference hg19 [https://www.10xgenomics.com/support/software/cell-ranger/downloads/cr-ref-build-steps]. The published data sets used for GSEA are available in the Molecular Signatures Database (MSigDB) [https://www.gsea-msigdb.org/gsea/msigdb/]: Hallmark (PMID: 10592173), Reactome (PMID: 29145629) and KEGG (PMID: 10592173); and in the GEO repository under accession number GSE120369 (from Stephenson et al.[35]. PMID: 30642980). Flow cytometry data files for the analysis of human bone marrow plasma cells are available in the Flow Repository under accession ID FR-FCM-Z7A5, while those for the analysis of the co-culture of helper T cells with memory B cells can be found under accession ID FR-FCM-Z7CB. All other data are available in the article and its Supplementary files or from the corresponding author upon request. Additional support for further analysis of the study findings is available upon request from the corresponding author or the special correspondence for bioinformatics [pawel.durek@drfz.de]. Source data are provided with this paper.

## Code availability

The software used in this study is open source. No custom code was used. Cellranger from 10x genomics: https://support.10xgenomics.com/single-cell-gene-expression/software/downloads/latest. Seurat packages 4.1.1: https://cloud.r-project.org/web/packages/Seurat/index.html. Monocle3 package: https://cole-trapnell-lab.github.io/monocle3/docs/installation/

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

## Acknowledgements

The authors are most grateful to the patients for their consent to participate in this study. We would like to thank Antje Blankenstein, Johanna Penzlin and Simon Reinke from the BCRT Cell Harvesting Laboratory and the Charité Orthopedics Surgery Department for the provision of samples. We also thank Jenny Kirsch and Toralf Kaiser from the DRFZ Flow Cytometry Core Facility for support in cell sorting and Mairi McGrath for text revision. This work was kindly supported by the following entities: the German Research Foundation (DFG) through the grants LI3540/1-1 to A.C.L., SCHR1658/1-1 to E.V.S., Do491/8-1/2 (SPP Immunobone) and Do491/7–5, 10-1, 11-1 to T.D., TRR130/TP24 to H.E.M. and T.D, TRR130/TP16 to A.R., HU1294/8-1/2 to A.H., and the Clinical Research Unit KFO 5023 'BecauseY'/Project number 504745852 to A.K.; the Federal Ministry of Education and Research (BMBF) with financing of the projects TReAT and CONAN to M.F.M., and the grant BCOVIT, 01KI20161 to E.V.S.; the state of Berlin and the "European Regional Development Fund" through the grant ERDF 2014-2020, EFRE 1.8/11, to M.F.M.; the Leibniz Association through the Leibniz Collaborative Excellence projects TarGArt to M.F.M., ImpACT to A.K. and M.F.M., and CHROQ to F.M.; the Berlin senate through the financing of the project: "Modulation of the mucosal immune response to prevent severe COVID-19 disease progression by commensal bacteria or vaccination" to M.F.M.; the Berlin Institute of Health through the Starting Grant Multi-Omics Characterisation of SARS-CoV-2 infection, Project 6 "Identifying immunological targets in COVID-19" to M.F.M.; COLCIENCIAS scholarship call 727-2015 to H.R-A.; and the Charité-Universitätsmedizin Berlin through the Rahel-Hirsch-Stipendium to A.L.S.

## Author contributions

Conceptualisation: A.R., T.D., M.F.M.; methodology: M.F.G., Y.C., P.D., H.R.A., L.B., F.S., G.M.G., A.C.L.; software: P.D., F.H., F.S.; validation: M.F.G., A.C.L., M.F.M.; formal analysis: P.D., F.H., F.S.; investigation: M.F.G., Y.C., H.R.A., L.B., G.M.G., A.L.S., A.N., A.W., M.B., J.C.R., K.L., S.H., V.D.D.; resources: S.H., C.H., Q.C., M.W., H.E.M., E.V.S.; data curation: M.F.G.; P.D., M.F.M.; writing–original draft: M.F.G., M.F.M.; writing–review & editing: M.F.G., Y.C., F.M., A.C.L., A.R., T.D.; visualisation: M.F.G., Y.C., P.D., F.H., F.S.; supervision: E.H., M.M., A.C.L., A.K., C.P., A.H., A.R., T.D., M.F.M.; project administration: T.D., M.F.M.; funding acquisition: A.R., T.D., M.F.M.

## Funding

## Competing interests

The authors declare no competing interests.

## Additional information

[1]Deutsches Rheuma-Forschungszentrum Berlin, ein Institut der Leibniz Gemeinschaft, Berlin, Germany. [2]Department of Rheumatology and Clinical Immunology, Charité-Universitätsmedizin Berlin, Berlin, Germany. [3]Grupo de Inmunología Celular e Inmunogenética, Facultad de Medicina, Instituto de Investigaciones Médicas, Universidad de Antioquia UdeA, Medellín, Colombia. [4]Department of Nephrology and Medical Intensive Care, Charité-Universitätsmedizin Berlin, corporate member of Freie Universität Berlin, Humboldt-Universität zu Berlin, and Berlin Institute of Health, Berlin, Germany. [5]Institute of Immunology, University Hospital Schleswig-Holstein, Kiel, Germany. [6]Department of Orthopedic Surgery, Charité-Universitätsmedizin Berlin, Berlin, Germany. [7]Paul-Ehrlich-Institut, Bundesinstitut für Impfstoffe und biomedizinische Arzneimittel, Langen, Germany. [8]Berlin Institute of Health at Charité-Universitätsmedizin Berlin, Berlin, Germany. [9]Department of Microbiology and Infection Immunology, Charité-Universitätsmedizin Berlin, Berlin, Germany. [10]These authors contributed equally: Marta Ferreira-Gomes, Yidan Chen, Pawel Durek. [11]These authors jointly supervised this work: Andreas Radbruch, Thomas Dörner, Mir-Farzin Mashreghi. ✉e-mail: mashreghi@drfz.de

