## [Peer Review File · Nature Communications]

Recruitment of plasma cells from IL21-dependent and -independent immune reactions to the bone marrowEditorial Note: This manuscript has been previously reviewed at another journal that is not operating a transparent peer review scheme. This document only contains reviewer comments and rebuttal letters for versions considered at *Nature Communications*. Mentions of the other journal have been redacted.

REVIEWER COMMENTS

Reviewer #1 (Remarks to the Author):

In this study, the authors present a detailed characterization of human bone marrow plasma cells (BMPC) from multiple donors using single-cell sequencing. Additionally, they employ clonal lineage analysis to trace the recruitment of antibody-secreting cells (ASC) in the blood as they exit immune reactions and migrate to the bone marrow. In the revised work, the authors conduct additional experiments using ex-vivo co-culture of helper T cells with memory B cells to investigate the expression of CD19 on activated cells. Their findings show that IL-21 suppresses CD19 expression in activated B cells. Based on these analyses, the authors propose that CD19^{low}-BMPCs are derived from follicular immune reactions, while CD19^{high}-BMPCs originate from extrafollicular immune reactions (such as the reactivation of memory B cells). The key message of this study is that primary immune responses result in plasma cells residing within both the CD19-low and CD19-high compartments (“clans”), whereas secondary immune responses primarily generate plasma cells mainly found in the CD19-high compartment.

In this revised manuscript, the authors have addressed our original concerns regarding different analyses, and new data are provided to support their key message. The paper merits publication in *Nature Communications*.

Response to a few minor comments is desirable, as noted below:

1. It is not clear why only FR regions were considered for the calculation of SHM. The FRs are usually less mutated (except BCRs from individuals with chronic HIV infection) and generally,

the mutations are calculated for the entire V region that includes both FRs and CDRs.

2. What are the relative sizes of individual antigen-specific clonal families generated in follicular and extrafollicular responses that are traced to BM after using the improved definition of clonotype?

3. Line 343: “..the first comprehensive single-cell gene. Expression analysis of human bone marrow plasma cells..” – the claim throughout the manuscript should be removed as there already is a study on the heterogeneity of human bone marrow plasma cells published by another group Duan et al., 2023 (PMID37355988).

Reviewer #2 (Remarks to the Author):

I very much enjoyed reading the manuscript of Ferreira-Gomes et al. It reflects a substantial body of work with careful and thorough analysis, including the scRNAseq, BCR analysis for inferred specificity, selection and clonality, and reconstitution of aspects of the response using ASC from blood after vaccination or from in vitro cultures. The associations the authors make between the vaccination and infection/clinical history of the individual donors and the BMPC are very interesting and will be a very useful addition to the understanding of the development of immunity. While I thought some of the associations were speculative, the authors were quite clear in what they had done, leaving it to the reader to decide whether they agreed or not with the argument of formative instruction being put forward. One of the other associations, however, I would suggest could use some support from the literature, if possible. This being the tenant that IL-21 is only involved in the differentiation of ASC from germinal center B cells. I am not aware of this being fact and can only find references that support the opposite assertion (that IL-21 is involved in extra-follicular ASC differentiation, including from memory B cells). This point needs clarification.

I read also the initial referee reports and the response of the authors and in my opinion, they have addressed in a substantial manner the criticisms that arose. I would have little hesitation in recommending this publication in Nat Com, as I think it will make a significant contribution to the understanding of long-lived antibody secreting plasma cells, both in their formation and in their persistence.

Ettinger et al., (2007) IL-21 and BAFF/BlyS synergize in stimulating plasma cell differentiation from a unique population of human splenic memory B cells. J Immunol. (2007) 178:2872–82

Wang, S. et al. (2018) IL-21 drives expansion and plasma cell differentiation of autoreactive CD11chi T-bet+ B cells in SLE. Nat. Commun. 9, 1758.

Reviewer #3 (Remarks to the Author):

The Ferreira-Gomes et. al. rebuttal to a [journal name redacted] submission is being considered for Nat Comm. The revised manuscript has significant changes with new emphasis of follicular and extrafollicular origins of BMPC CD19hi and CD19lo. The writing is improved and the authors have address many concerns. However, The manuscript provides a lot of datasets that would be interest to scientific community but their conclusions are overreaching and speculative with experiments that do not support these conclusions. The concern is in the analyses of how blood ASC have origins of BMPC through the transcriptome profiles when they are highly different as the authors themselves point out. Definitions of public clones are also not clarified in this revision. Despite the important datasets, the main conclusions of IL-21 mediating follicular vs extrafollicular conclusions are not supported. Thus, raising the following concerns.

1. As they applied Q/C filters on the original 38,235 single cell transcriptomes, they removed many that were B cells. What is the final number of BMPC analyzed (which must have been less for BMPC). Additionally, it is good to see 48,000 mean reads per cell but with some BMPC with 80-90% of the transcriptome that are Ig genes, they may be underrepresenting these BMPC subsets. The reclustering do not remove the low quality cells and B cells.
2. Their own data does not support the conclusions of primary immunization vs secondary immunizations and location in the BMPC. If TT-specific BMPC are found in all compartments due to a combination of primary and secondary immune responses but spike-specific BMPC are new antigens and they too are also filling all BM compartments and so it is unclear how

this model shows that secondary immunizations fill mainly the CD19hi compartment (fig 2e and supplemental fig 2e, supplemental tables) in nearly all donors. Interestingly, sub 555 and 556 who had been vaccinated 3 times with their recent dose within 71d did not have clones in the CD19hi (clan 0) compartments. Thus, their own datasets are not conclusive to support that primary immune responses reside only CD19lo and CD19hi BMPC compartments while secondary responses fill the CD19hi BMPC clusters.

3. There are concerns about the definition of “public clones” and the data presented in the tables. They define that “based on CDR3 similarity, we designated those clones with more than 80% similarity as “public” RBD/spike- and TT-specific clones” (line 187). Public clones are defined as a clonotype that is shared by at least two different individuals. It is not clear how the authors are using the definition of public clones. It seems that the authors are using the definition of a public clone as a BCR sequence that is shared between blood and BMPC from the same individual. Then, “of the 38,235 BMPC analysed (likely less which are ASC), there are 84 expressed spike-specific and 120 tetanus-specific public clonotypes (line 189). Again, it is not clear if all 84 clones are just the public clones or the total number of spike-specific clones found in the BMPC. If these were indeed 84 public clones within the 8 individuals, what was the total number of spike/RBD- specific clones? Based on frequency of antigen-specific BMPC clones, it would seem reasonable to have $84/38235$ BM cells or approximately 0.2% that would spike specific thus it is likely that these were the total of spike specific BMPC clones that matched from the memory B cell sequences. Similarly, $120/38,235$ would be 0.3% of ASC are TT-specific BMPC. From the data in the tables, these sequences do not appear to be 84 individual public clonotypes but RBD/spike specific BMPC clones identified from memory B cells (BMPC_putative_RBD_spike_clones tab in table 434723_0_data_set) since line 43, 44, 47, 50, 51, 53 are from the same clonal origin based on the V, J and CDR3. Also, 43 and 53 are identical clones in different BMPC clans, thus this list is not 84 individual public clones. Again, this is likely a total of spike specific BMPC clones. To identify true “public” clones, they should align the same clonotypes from different individuals to demonstrate this.

Where is table 1 with the 8 individuals that provide the dataset for the scRNAseq of the BMPC? Is it labeled 434723_0_data set_?

4. The CD19hi and the CD19lo is relative to ASC. Many papers have shown that CD19 expression is downregulated after a secondary vaccination (Tooze et al JI, Joyner LSA 2022 show decrease CD19 in cultures). However, it is hard to interpret the analysis of in vitro B cell differentiated cultures and evaluation of differences in transcriptomes then to project them onto the BMPC (fig 4a,b). For example, identifying B cell cultures with high CD19 expression after 12 h and then projecting them onto the ASC cultures do not suggest that these B cells are the origins of the CD19hi BMPC. Thus, it is difficult to draw conclusions from the data presented in figure 4a,b.

5. There is concern with projecting blood ASC transcriptomes onto BMPC is the analysis and the conclusions that are drawn based on these analyses which are likely hypothesis generating and not conclusive.

6. A reclustered UMAP of the blood and BM scRNAseq is important but again not shown. In the authors' reply they state that "if the blood and BMPC are analyzed in the same UMAP, they would cluster separately". This is very important. If the BMPC and the blood ASC separate, they must have major differences in transcriptomes and thus require experimental validation that they a blood ASC eventually becomes a specific BMPC clan. The analysis methods of using GSEA on the single cell level from the different blood ASC and then matching the differential signatures to the BMPC signatures to define the hypothetical allocation of ASC to the BMPC clans is just speculative and are not conclusive.

7. They authors also conclude that the primary immunization leads to filling the BMPC clan 1(CD19lo) and clan 0 (CD19hi) and secondary immunisation fills the CD19hi clans (clan0). However, this is never proven. If the primary immunisation shows filling both of these both clans, how can using a single time point of BMPC demonstrate that secondary immunisation fills only the CD19hi clan. How can one know if these clans were filled during the primary or secondary immunisation. Again, these conclusions are mere speculation.

8. Supplemental figure 5a only shows only the blood ASC of different vaccine responses. It is really important to look at the differences between the transcriptomes of blood and BMPC

and recluster among blood ASC and to project onto BMPC.

9. It is difficult to draw conclusions from the interpretation of the projected transcriptomes of B cell 12-24h after different stimulation conditions onto BMPC (fig 4a, b). Which B cell were used but more importantly it is difficult to interpret the meaning of projecting transcriptomes of 12-24h of B cell stimulated cultures which would be proliferating B cells onto BMPC which are likely to be vastly different. Thus, projecting differences in B cell cultures after 12-24h based on selected cytokines onto “signatures by GSEA onto the UMAP of peripheral ASC collected at different times post-vaccination” does not prove that these B cell will fill these BMPC compartments. This is again speculation and do not confirm the B cell assignment of BMPC clans.

10. In Fig 4c flow data of Bmem cultures with Tfh, they use memory B cells isolated from the tonsils to attempt to demonstrate the role of IL-21 downregulating CD19 expression on B cells activated to differentiate into ASC. Usually naive and GC B cells express IL-21R and only with activation (i.e.anti-CD40) do memory B cells upregulate IL-21R, thus the use of B cells with high IL-21R would have been more informative. Also, many Bmem cells isolated from the tonsil (can be GC derived) making these experiments difficult to analyze. Furthermore, it is unclear if the starting Bmem cells could have been contaminated with recently generated ASC at the start of the cultures leading to such variability. Interestingly, the in vitro Bmem cells cultures had such variability (N=2) of CD19hi ASC without IL-21 (only 12 and 45%) thus making it difficult to assess the downregulation due to IL-21 alone. Even if IL-21 could downregulates CD19 expression in the cultures, it is unclear the level of IL-21R on these Bmem cells at the start. Hence, it is difficult to conclude the mechanism of CD19 downregulation of BMPC in vivo with these experiments.

11. IL-21 can also be derived from extrafollicular T cells (Odegard JM et al JEM 2008) and thus it is difficult to conclude that the role of IL-21 in follicular (not extrafollicular) immune responses which is their main conclusion. The conclusion of follicular (IL21 mediated) vs extrafollicular responses resulting in variable CD19 expression based on fig 2 and 4 are just speculative without validation. There are no LN bx with BMPC from the same individual to demonstrate follicular and extrafollicular B cell origins of BMPC. Even if we accept their interpretation of clan 0 and 1 as CD19hi and CD19lo respectively, the authors do not

demonstrate the “recruitment of plasma cells from follicular and extrafollicular immune reactions to the BM”. They attribute the importance of IL-21 suppressing CD19 expression on follicularly derived B cells but their data do not support these conclusions.

Point-by-point response to the reviewers' comments

Reviewer #1 (Remarks to the Author):

In this study, the authors present a detailed characterization of human bone marrow plasma cells (BMPC) from multiple donors using single-cell sequencing. Additionally, they employ clonal lineage analysis to trace the recruitment of antibody-secreting cells (ASC) in the blood as they exit immune reactions and migrate to the bone marrow. In the revised work, the authors conduct additional experiments using ex-vivo co-culture of helper T cells with memory B cells to investigate the expression of CD19 on activated cells. Their findings show that IL-21 suppresses CD19 expression in activated B cells. Based on these analyses, the authors propose that CD19^{low}-BMPCs are derived from follicular immune reactions, while CD19^{high}-BMPCs originate from extrafollicular immune reactions (such as the reactivation of memory B cells). The key message of this study is that primary immune responses result in plasma cells residing within both the CD19-low and CD19-high compartments ("clans"), whereas secondary immune responses primarily generate plasma cells mainly found in the CD19-high compartment.

In this revised manuscript, the authors have addressed our original concerns regarding different analyses, and new data are provided to support their key message. The paper merits publication in Nature Communications.

Response to a few minor comments is desirable, as noted below:

1. It is not clear why only FR regions were considered for the calculation of SHM. The FRs are usually less mutated (except BCRs from individuals with chronic HIV infection) and generally, the mutations are calculated for the entire V region that includes both FRs and CDRs.

We thank the reviewer for this comment. We have now adjusted the figures (Figures 2b, c, d and g, Supplementary Figure 2c and Figure 3d) so that mutations in both Framework and CDR regions are presented, i.e a complete view on SHM. We have also changed the text accordingly (lines 170 to 173 and 197 to 202).

2. What are the relative sizes of individual antigen-specific clonal families generated in follicular and extrafollicular responses that are traced to BM after using the improved definition of clonotype?

Antigen-specific plasma cells expressing "public clone antibodies" among all BM plasma cells make up 0.05 to 0.32% for spike- and 0 to 2.03% for tetanus-specific clones among CD19^{high} BMPC, and 0.04 to 0.24% for spike- and 0 to 1.25% for tetanus-specific clones among the CD19^{low} BMPC. This information is now included in the revised version of the manuscript (lines 193 to 197).

3. Line 343: "...the first comprehensive single-cell gene. Expression analysis of human bone marrow plasma cells.." – the claim throughout the manuscript should be removed as there already is a study on the heterogeneity of human bone marrow plasma cells published by another group Duan et al., 2023 (PMID37355988).

We have now removed this claim in the revised version of our manuscript, and cited the work of Duan and colleagues (lines 396 to 397).

Reviewer #2 (Remarks to the Author):

I very much enjoyed reading the manuscript of Ferreira-Gomes et al. It reflects a substantial body of

work with careful and thorough analysis, including the scRNAseq, BCR analysis for inferred specificity, selection and clonality, and reconstitution of aspects of the response using ASC from blood after vaccination or from in vitro cultures. The associations the authors make between the vaccination and infection/clinical history of the individual donors and the BMPC are very interesting and will be a very useful addition to the understanding of the development of immunity. While I thought some of the associations were speculative, the authors were quite clear in what they had done, leaving it to the reader to decide whether they agreed or not with the argument of formative instruction being put forward. One of the other associations, however, I would suggest could use some support from the literature, if possible. This being the tenant that IL-21 is only involved in the differentiation of ASC from germinal center B cells. I am not aware of this being fact and can only find references that support the opposite assertion (that IL-21 is involved in extra-follicular ASC differentiation, including from memory B cells). This point needs clarification.

I read also the initial referee reports and the response of the authors and in my opinion, they have addressed in a substantial manner the criticisms that arose. I would have little hesitation in recommending this publication in Nat Com, as I think it will make a significant contribution to the understanding of long-lived antibody secreting plasma cells, both in their formation and in their persistence.

Ettinger et al., (2007) IL-21 and BAFF/BLyS synergize in stimulating plasma cell differentiation from a unique population of human splenic memory B cells. J Immunol. (2007) 178:2872–82

Wang, S. et al. (2018) IL-21 drives expansion and plasma cell differentiation of autoreactive CD11chi T-bet+ B cells in SLE. Nat. Commun. 9, 1758.

This is an interesting point and fun to discuss. Our data show that IL-21 does suppress expression of CD19 on activated B cells. We consider this already a major and original advance in the understanding of BMPC heterogeneity and derivation. We then show that BMPC from original antigenic stimulations are both CD19 low or high, while those of repeated antigenic stimulation are almost exclusively CD19high, i.e. those BMPC are derived from activated B cells which did not see IL-21. It is probably undisputed, and we provide citations, that IL-21 is an essential component of follicular B cell activation. Whether or not it is involved in extrafollicular B cell activation, could be discussed, and the references cited by the referee (now mentioned in the revised version of the manuscript, line 452) suggest that under certain circumstances IL-21 may contribute to extrafollicular B cell activation. However, not in the situations we have analysed here. In that respect, our data really contribute to clarify that in extrafollicular reactivations like the ones we have analysed, IL-21 does not play a role.

Reviewer #3 (Remarks to the Author):

The Ferreira-Gomes et. al. rebuttal to a [journal name redacted] submission is being considered for Nat Comm. The revised manuscript has significant changes with new emphasis of follicular and extrafollicular origins of BMPC CD19hi and CD19lo. The writing is improved and the authors have address many concerns. However, The manuscript provides a lot of datasets that would be interest to scientific community but their conclusions are overreaching and speculative with experiments that do not support these conclusions. The concern is in the analyses of how blood ASC have origins of BMPC through the transcriptome profiles when they are highly different as the authors themselves point out. Definitions of public clones are also not clarified in this revision. Despite the important datasets, the main conclusions of IL-21 mediating follicular vs extrafollicular conclusions are not supported. Thus, raising the following concerns.

1. As they applied Q/C filters on the original 38,235 single cell transcriptomes, they removed many

that were B cells. What is the final number of BMPC analyzed (which must have been less for BMPC). Additionally, it is good to see 48,000 mean reads per cell but with some BMPC with 80-90% of the transcriptome that are Ig genes, they may be underrepresenting these BMPC subsets. The reclustering do not remove the low quality cells and B cells.

We are sorry for the confusion. As stated in the figure legend (lines 1140 to 1142), 38235 is the number of BMPC after removal of contaminant B cells and lower quality cells, while the original number of single cell transcriptomes obtained is 49347 (Supplementary Figure 1a line 1273 and Methods line 591). To make these numbers clearer, we have now also stated that in the text of the manuscript (lines 102 to 103 and 115 to 119).

The mean number of reads per cell is 48000, more than twice as many as recommended and normally used. This should account for the about 50% of reads representing antibody gene expression. It does, since the median number of genes per cell captured is more than 1700, a number exceeding by far the widely used threshold of 1000 genes per cell for cells of acceptable quality. The low quality cells were defined according to low number of genes captured per cell, as well as high mitochondrial gene expression. In addition, the fact that the BCR transcripts of these cells were not captured by BCR-Profiling, is taken as an additional evidence of damaged cells. This information is now added in the new version of Supplementary Figure 1b in the revised manuscript. We are confident, that our data provide a true picture of BMPC at the single cell level. This is also evident from the expression of generic plasma cell genes, like PRDM1 and SDC1, by the cells analysed. Regarding the reclustering figure (Reviewer Figure 1), we apologise for having the wrong cluster labelling in Figure 1a. As it was correctly described in the figure legend, the reclustering only included BMPC after exclusion of contaminant and poor quality cells. These cells are the same 38235 cells analysed in Figure 1 of the manuscript. We are submitting the revised Reviewer Figure 1 with the correct cluster labelling.

2. Their own data does not support the conclusions of primary immunization vs secondary immunizations and location in the BMPC. If TT-specific BMPC are found in all compartments due to a combination of primary and secondary immune responses but spike-specific BMPC are new antigens and they too are also filling all BM compartments and so it is unclear how this model shows that secondary immunizations fill mainly the CD19hi compartment (fig 2e and supplemental fig 2e, supplemental tables) in nearly all donors. Interestingly, sub 555 and 556 who had been vaccinated 3 times with their recent dose within 71d did not have clones in the CD19hi (clan 0) compartments. Thus, their own datasets are not conclusive to support that primary immune responses reside only CD19lo and CD19hi BMPC compartments while secondary responses fill the CD19hi BMPC clusters.

Figure 2e shows an overlay of antigen-specific public clones of BMPC, representing 8 donors each. We have now changed Supplementary Figure 2e to show the cluster distribution in all donors individually. Although we do not exactly know how many contacts each individual had with the respective antigen, all had had multiple contacts with Spike and all but one also with tetanus. We did not compare persons with few spike versus persons with many tetanus contacts. Our results are in accordance with the notion that repeated antigen contact leads to deposition of antigen-specific BMPC in most clans.

With respect to subs 555 and 556, the new Supplementary Figure 2e to makes now clear that both of them have CD19^{high} spike-specific BMPC.

3. There are concerns about the definition of “public clones” and the data presented in the tables. They define that “based on CDR3 similarity, we designated those clones with more than 80% similarity as “public” RBD/spike- and TT-specific clones” (line 187). Public clones are defined as a clonotype that is shared by at least two different individuals. It is not clear how the authors are using the definition of public clones. It seems that the authors are using the definition of a public clone as a BCR sequence that is shared between blood and BMPC from the same individual. Then, “of the 38,235 BMPC analysed (likely less which are ASC), there are 84 expressed spike-specific and 120 tetanus-

specific public clonotypes (line 189). Again, it is not clear if all 84 clones are just the public clones or the total number of spike-specific clones found in the BMPC. If these were indeed 84 public clones within the 8 individuals, what was the total number of spike/RBD- specific clones? Based on frequency of antigen-specific BMPC clones, it would seem reasonable to have 84/38235 BM cells or approximately 0.2% that would spike specific thus it is likely that these were the total of spike specific BMPC clones that matched from the memory B cell sequences. Similarly, 120/38,235 would be 0.3% of ASC are TT-specific BMPC. From the data in the tables, these sequences do not appear to be 84 individual public clonotypes but RBD/spike specific BMPC clones identified from memory B cells (BMPC_putative_RBD_spike_clones tab in table 434723_0_data_set) since line 43, 44, 47, 50, 51, 53 are from the same clonal origin based on the V, J and CDR3. Also, 43 and 53 are identical clones in different BMPC clans, thus this list is not 84 individual public clones. Again, this is likely a total of spike specific BMPC clones. To identify true “public” clones, they should align the same clonotypes from different individuals to demonstrate this.

We agree with the reviewer that “public clone” defines a clone that is shared between different individuals, and have now clearly stated that in the text (lines 189 to 189). 84 is indeed the number of total spike-specific “public” clones among BMPC, as listed in tab BMPC_putative_RBD_spike_clones.

Where is table 1 with the 8 individuals that provide the dataset for the scRNAseq of the BMPC? Is it labeled 434723_0_data set_?

The table providing information on the 8 individuals analysed by single cell sequencing is provided as Supplementary Table1.

4. The CD19^{hi} and the CD19^{lo} is relative to ASC. Many papers have shown that CD19 expression is downregulated after a secondary vaccination (Tooze et al JI, Joyner LSA 2022 show decrease CD19 in cultures). However, it is hard to interpret the analysis of in vitro B cell differentiated cultures and evaluation of differences in transcriptomes then to project them onto the BMPC (fig 4a,b). For example, identifying B cell cultures with high CD19 expression after 12 h and then projecting them onto the ASC cultures do not suggest that these B cells are the origins of the CD19^{hi} BMPC. Thus, it is difficult to draw conclusions from the data presented in figure 4a,b.

CD19 expression may indeed be downregulated after secondary vaccination in some activated cells. However, in the repeated antigen contacts analysed here, as shown in Figure 3 in the end by far the most cells are located to CD19^{high} clans. We do not use Figures 4a and b to conclude that IL-21 downregulates CD19 expression. This claim is based on Figures 4c and d. Figures 4a and b, together with Figures 1e and g, show that different clans represent BMPC coming from immune reactions with different instructive cytokines involved.

5. There is concern with projecting blood ASC transcriptomes onto BMPC is the analysis and the conclusions that are drawn based on these analyses which are likely hypothesis generating and not conclusive.

In Figure 3c we show that the circulating ASC at the time points we had purposively chosen to analyse are specific for the antigen. We then projected their gene expression pattern (gene set) onto BMPC and identified BMPC with a similar gene expression profile. We consider this a fair, conclusive and transparent way to identify BMPC generated in a particular immune reaction. It generates a new hypothesis, which is of interest for the community. Hypotheses based on data as we provide them here, are the essence of science, and not just speculation. Challenges welcome.

6. A reclustered UMAP of the blood and BM scRNAseq is important but again not shown. In the authors' reply they state that “if the blood and BMPC are analyzed in the same UMAP, they would cluster separately”. This is very important. If the BMPC and the blood ASC separate, they must have

major differences in transcriptomes and thus require experimental validation that they a blood ASC eventually becomes a specific BMPC clan. The analysis methods of using GSEA on the single cell level from the different blood ASC and then matching the differential signatures to the BMPC signatures to define the hypothetical allocation of ASC to the BMPC clans is just speculative and are not conclusive.

In a simple UMAP circulating ASC and BMPC would cluster separately, since they differ e.g. in genes of proliferation and mobility. This has been shown before (Joyner *et al.*, doi: 10.26508/lsa.202101285). In a UMAP based on an integration approach using anchor genes to integrate transcriptomes, BMPC cluster with circulating ASC. This is shown in the new Supplementary Figure 3e. Thus, we consider our allocations to be conclusive.

7. They authors also conclude that the primary immunization leads to filling the BMPC clan 1(CD19lo) and clan 0 (CD19hi) and secondary immunisation fills the CD19hi clans (clan0). However, this is never proven. If the primary immunisation shows filling both of these both clans, how can using a single time point of BMPC demonstrate that secondary immunisation fills only the CD19hi clan. How can one know if these clans were filled during the primary or secondary immunisation. Again, these conclusions are mere speculation.

We kindly disagree. We use differential gene sets expressed by circulating ASC, generated in a defined immune reaction, to identify the BMPC expressing these gene sets. Thus we allocate the ASC generated to their BMPC cousins. Why should not a BMPC clan be filled both in primary and later also in secondary immune reactions? This is what we show for the CD19^{high} clan 0. Based on the arguments provided above, in response to points 5 and 6, we consider our conclusions to be conclusive.

8. Supplemental figure 5a only shows only the blood ASC of different vaccine responses. It is really important to look at the differences between the transcriptomes of blood and BMPC and recluster among blood ASC and to project onto BMPC.

Supplementary Figure 5 illustrates the differential gene sets defining the signatures of circulating ASC. In the new Supplementary Figure 3e we now show the integrated UMAP clustering of these ASC to the BMPC.

9. It is difficult to draw conclusions from the interpretation of the projected transcriptomes of B cell 12-24h after different stimulation conditions onto BMPC (fig 4a, b). Which B cell were used but more importantly it is difficult to interpret the meaning of projecting transcriptomes of 12-24h of B cell stimulated cultures which would be proliferating B cells onto BMPC which are likely to be vastly different. Thus, projecting differences in B cell cultures after 12-24h based on selected cytokines onto "signatures by GSEA onto the UMAP of peripheral ASC collected at different times post-vaccination" does not prove that these B cell will fill these BMPC compartments. This is again speculation and do not confirm the B cell assignment of BMPC clans.

This point is essentially the same as the raised point 4. We have responded to it above. We simply ask to respect that we transparently use gene sets of circulating ASC to identify their BMPC cousins, and gene sets of *ex vivo* activated B cells to identify their imprinting instructions.

10. In Fig 4c flow data of Bmem cultures with Tfh, they use memory B cells isolated from the tonsils to attempt to demonstrate the role of IL-21 downregulating CD19 expression on B cells activated to differentiate into ASC. Usually naive and GC B cells express IL-21R and only with activation (i.e. anti-CD40) do memory B cells upregulate IL-21R, thus the use of B cells with high IL-21R would have been more informative. Also, many Bmem cells isolated from the tonsil (can be GC derived) making these experiments difficult to analyze. Furthermore, it is unclear if the starting Bmem cells could have been contaminated with recently generated ASC at the start of the cultures leading to such variability.

Interestingly, the in vitro Bmem cells cultures had such variability (N=2) of CD19^{hi} ASC without IL-21 (only 12 and 45%) thus making it difficult to assess the downregulation due to IL-21 alone. Even if IL-21 could downregulate CD19 expression in the cultures, it is unclear the level of IL-21R on these Bmem cells at the start. Hence, it is difficult to conclude the mechanism of CD19 downregulation of BMPC in vivo with these experiments.

We do not show the mechanism of CD19 downregulation by IL-21, but rather provide the original evidence that IL-21 indeed can downregulate CD19 expression. This alone is an interesting finding, given the state-of-the-art confusion about the biological basis of differential CD19 expression by BMPC. With this experiment we show that mimicking of germinal centre conditions, even when using lower IL-21R expressing cells, leads to the downregulation of CD19 in ASC. Despite the variability, all 4 donors show a marked downregulation of CD19 which is blocked by soluble IL-21R, being in our eyes a clear sign of IL-21-mediated downregulation of CD19 in plasmablasts. Contamination of memory B cells with ASC was excluded by after sort purity check.

11. IL-21 can also be derived from extrafollicular T cells (Odegard JM et al JEM 2008) and thus it is difficult to conclude that the role of IL-21 in follicular (not extrafollicular) immune responses which is their main conclusion. The conclusion of follicular (IL21 mediated) vs extrafollicular responses resulting in variable CD19 expression based on fig 2 and 4 are just speculative without validation. There are no LN bx with BMPC from the same individual to demonstrate follicular and extrafollicular B cell origins of BMPC. Even if we accept their interpretation of clan 0 and 1 as CD19^{hi} and CD19^{lo} respectively, the authors do not demonstrate the “recruitment of plasma cells from follicular and extrafollicular immune reactions to the BM”. They attribute the importance of IL-21 suppressing CD19 expression on follicularly derived B cells but their data do not support these conclusions.

We kindly disagree. See also our response to reviewer #2, above. The conclusion of our analysis is that in primary B cell activations, both CD19^{low} and CD19^{high} BMPC are generated, while in secondary activations it is mostly CD19^{high} BMPC. The question whether IL-21 also may contribute to some extrafollicular reactions is irrelevant with respect to our data. We can safely conclude that those CD19^{high} BMPC by us analysed did not see IL-21 and thus are from extrafollicular reactivations. Whether in the primary activations BMPC are generated only in follicular or also in extrafollicular reactions, is less clear. In summary, we think that for the immune reactions analysed here, our data solidly support our claims.

REVIEWER COMMENTS

Reviewer #1 (Remarks to the Author):

The paper has been extensively revised and is acceptable for publication.

Reviewer #2 (Remarks to the Author):

The work is acceptable for publication.

Reviewer #3 (Remarks to the Author):

The authors have addressed some of these questions but they still have not definitively proven the main conclusion in the title that plasma cells are recruited from follicular or extrafollicular reactions. The data for the public clones have not been clarified and they need to provide the dataset and the VDJ sequences. They also need to state the conclusions as speculative and not conclusive since they do not provide evidence that the blood ASC actually fill a particular BMPC compartment.

I would like to see this work published because there is interesting data presented; however, in its current form the conclusions are too speculative and there are several things that need to be changed: (1) the title of the “extrafollicular vs follicular” claims, (2) the need to provide datasets of the public clones with the alignments from multiple individuals, and (3) the claims of the blood ASC and BM ASC transcriptome overlays that do not overlap by UMAP to conclude that their “integrative approach with anchor genes” provides enough evidence that these blood ASC align with CD19^{hi} and CD19^{lo} BMPC compartments and so must eventually become these BMPC.

1. They need to remove the “follicular and extrafollicular” in the title of the manuscript because the data do not support this conclusion. Their data show that the location of the putative spike and tetanus-specific ASC in the BMPC compartments which is found in all BMPC compartments. This is interesting as it is.

2. (previous point #3) They need to report the CDR3 and VDJ sequences identified from the TT- and RBD/spike-specific memory B cells clones. Then, they need to report the BMPC clones that lined up with the TT- and the RBD/spike- specific memory B cells clones. How many total TT- and RBD/spike-specific BMPC clones lined up with the memory B cell clones? It appears that they provide only the number of “public clones” from the TT- and RBD/spike BMPC in the current dataset 434723_1_data_set_8110896_s1l1j (tab BMPC_putative RBD_spike_clones and BM_putative_TT_clones). This list of “public clones” in the BMPC that are putative spike-specific clones 84 and the tetanus-specific clones are 120. From this dataset, they need to provide the alignment of the VDJ and CDR3 sequence from the multiple donors to show that they are indeed “public clones”. This dataset does not show any Also, they need to show the number of private clones within the putative TT-specific and RBD/spike-specific clones that were not shared among the individuals. This data should be readily available.

3. They also report 749 spike-specific clonotypes among blood ASC at the peak of vaccination but these are antigen-enriched but not known to be antigen-specific. How did they show that these were “spike-specific clonotypes”? If they were from the total blood ASC although enriched but not known to be spike specific, then the term “spike specific” should be removed or written as “putative spike-specific Ig”. Since they are attempting to demonstrate clonal relationships among the clones, they need to show how these were antigen-specific to call them spike-specific especially if they will be calling them spike-specific Ig public clones (as in figure 3c-e). For clarity, what was the total number of ASC clones identified from the 36 individuals? Since antigen-specific ASC are enriched in the blood ASC after vaccination, they should be oligoclonal and thus there should be many that are identical or in the same clonal family. How many clonal families (clonotypes) were there? They should report total number and show the sequences of the individual clones and the same clonal family per individual (using their definition of same VJ same CDR3 length and 80% homology of the CDR3). They should report how they determined antigen-specificity (if by generating monoclonal antibodies). Finally, they also report that these are public clones (figure 3c, d) and so they need to show the alignment of the VDJ sequences from different individuals to demonstrate the “public clone”. Finally, from this list, how many were private clones (not shared among individuals) and how many were public clones

(shared between individuals). The alignments should be shown to illustrate public clones which are shared among individuals. If this were true that they found 749 public clonotypes, this would be a remarkable finding and needs to be shown clearly. The data of public RBD public clones are listed in the dataset 434723_1_data_set_8110896_s1l1j tab putative ASC_putative_RBD_spike_clones. In the dataset, there are very few that are shared among the individuals.

This information was previously requested to understand their datasets but they only change the text in the manuscript page line 191 to say that the 84 and 120 clones listed are “shared between individuals”. More data and clarification are required.

4. (previous point #5) The authors themselves report that the blood ASC and BMPC transcriptomes do not overlap when they are evaluated by simple UMAP analysis. This statement needs to be included in the paper because it is misleading to conclude that overlaying the blood ASC onto the BMPC transcriptomes using their integrated analysis proves that these are the same cell transcriptionally. They report that the UMAP analysis shows that they are very different and do not overlap; thus they cannot conclude that they are the same cells. However, they can report that they tried to suggest where these blood ASC may eventually reside in the BMPC by “using our integrated approach with anchor genes” in order to overlay the blood ASC onto the BMPC. Transcriptional analysis merely suggests and is hypothesis driving but they do not prove that certain blood ASC eventually reside in a particular BMPC compartment. For example, they do not time-stamp these cells or label them and follow them into the BMPC compartments. Thus, their conclusions are speculative and should be reported as such. This is human data and some experiments are limiting but there is still value in reporting as such but the interpretation should be reported appropriately so they do not make overarching conclusions from the dataset that cannot be proven.

They also say that blood ASC are different essentially due to “proliferation and mobility” genes and so they should remove the “proliferation and mobility” genes and show that they contain the similar transcriptomes as the BMPC if this is the only difference. This additional

analysis would be highly welcome.

5. (previous point #7). It is hard to conclude that primary immunization fills the BMPC clan 1 (CD19lo) and clan 0 (CD19hi) and secondary immunization fills the CD19hi or (clan 0). Interestingly, spike fills the CD19lo and CD19hi subsets with primary COVID-19 immunization but not in all patients as pointed out with subjects 555 and 556. Perhaps reporting that only 6/8 patients reported were demonstrate this finding? But now, the authors report that CD19lo subsets include clans 1, 4, 5, 6 to show demonstrate their point. If this is the case, why was this information not described previously. Now CD19hi clans now include 0, 8, 9, 11, 12, 13 and CD19lo included clans 1, 4, 5, and 6. If this is the case, they need to include this in the beginning of the description of the different clans. They probably need to expand supplementary figure 1e to include all the clans in the CD19hi and CD19lo groups.

6. They conclude that their dataset shows that primary responses fill CD19lo and CD19hi while secondary responses only fill the CD19hi compartment. However, looking in the BMPC primary and secondary would look the same because one could not distinguish the pre-existing old PC (which should be in both CD19lo and CD19hi) vs new arrivals into the BMPC (CD19hi) by antigen-specificity alone. Thus, it is unclear how they can make this conclusion. Also, the BM samples were obtained from individuals with COVID vaccines 3 times and unknown in two patients of primary vs secondary responses. A single BMPC with antigen-specificity cannot separate old existing PC vs new arrivals in the BMPC with antigen-specificity alone. However, the authors are using their integrated approaches with the blood and BM ASC transcriptome overlay to draw this conclusion. The problem is that the blood ASC and BMPC transcriptomes do not overlap on the UMAP. Their integrative approach is speculative, interesting, and intriguing but not conclusive that repeat priming does not replenish the CD19lo compartments. They can only hypothesize and require further experiments for testing.

Finally, in new figure after 3 doses of the vaccine (secondary response), they report higher numbers of CD19lo antigen-specific ASC which is not consistent with their model. If with

more 3 doses of the vaccine and increased time, how do they explain higher numbers of CD19lo BMPC? Additional clarity is needed to interpret this dataset.

Point-by-point response to the reviewers' comments

Reviewer #1 (Remarks to the Author):

The paper has been extensively revised and is acceptable for publication.

We thank the reviewer.

Reviewer #2 (Remarks to the Author):

The work is acceptable for publication.

We thank the reviewer.

Reviewer #3 (Remarks to the Author):

The authors have addressed some of these questions but they still have not definitively proven the main conclusion in the title that plasma cells are recruited from follicular or extrafollicular reactions. The data for the public clones have not been clarified and they need to provide the dataset and the VDJ sequences. They also need to state the conclusions as speculative and not conclusive since they do not provide evidence that the blood ASC actually fill a particular BMPC compartment.

I would like to see this work published because there is interesting data presented; however, in its current form the conclusions are too speculative and there are several things that need to be changed: (1) the title of the "extrafollicular vs follicular" claims, (2) the need to provide datasets of the public clones with the alignments from multiple individuals, and (3) the claims of the blood ASC and BM ASC transcriptome overlays that do not overlap by UMAP to conclude that their "integrative approach with anchor genes" provides enough evidence that these blood ASC align with CD19hi and CD19lo BMPC compartments and so must eventually become these BMPC.

The sequencing data used in this project, including the entire VDJ sequences, is being deposited in GEO. We would like to additionally point out that the VDJ sequences of the identified antigen-specific public clones were already available as Supplementary Data.

1. They need to remove the "follicular and extrafollicular" in the title of the manuscript because the data do not support this conclusion. Their data show that the location of the putative spike and tetanus-specific ASC in the BMPC compartments which is found in all BMPC compartments. This is interesting as it is.

We have changed the title to be more specific: "Recruitment of plasma cells from IL-21-dependent and -independent immune reactions to the bone marrow".

2. (previous point #3) They need to report the CDR3 and VDJ sequences identified from the TT- and RBD/spike-specific memory B cells clones. Then, they need to report the BMPC clones that lined up with the TT- and the RBD/spike- specific memory B cells clones. How many total TT- and RBC/spike-specific BMPC clones lined up with the memory B cell clones? It appears that they provide only the number of "public clones" from the TT- and RBD/spike BMPC in the current dataset 434723_1_data_set_8110896_s11l1j (tab BMPC_putative RBD_spike_clones and BM_putative_TT_clones). This list of "public clones" in the BMPC that are putative spike-specific clones

84 and the tetanus-specific clones are 120. From this dataset, they need to provide the alignment of the VDJ and CDR3 sequence from the multiple donors to show that they are indeed “public clones”. This dataset does not show any. Also, they need to show the number of private clones within the putative TT-specific and RBD/spike-specific clones that were not shared among the individuals. This data should be readily available.

The CDR3 sequences identified from TT- and RBD/spike-specific memory B cells clones are now included in the Supplementary Data (sheet: Sequence_Alignment_RBD_spike; Sequence_Alignment_TT). In these new tables the listing of the putative RBD/spike- or TT-specific clones have been reorganized to visualize alignment of the shared clones (see page 32, lines 1452 to 1467). This new presentation clarifies the identification of public clones, i.e. 84 (spike)/120 (TT) clones of antigen-specific memory B cells.

3. They also report 749 spike-specific clonotypes among blood ASC at the peak of vaccination but these are antigen-enriched but not known to be antigen-specific. How did they show that these were “spike-specific clonotypes”? If they were from the total blood ASC although enriched but not known to be spike specific, then the term “spike specific” should be removed or written as “putative spike-specific Ig”. Since they are attempting to demonstrate clonal relationships among the clones, they need to show how these were antigen-specific to call them spike-specific especially if they will be calling them spike-specific Ig public clones (as in figure 3c-e). For clarity, what was the total number of ASC clones identified from the 36 individuals? Since antigen-specific ASC are enriched in the blood ASC after vaccination, they should be oligoclonal and thus there should be many that are identical or in the same clonal family. How many clonal families (clonotypes) were there? They should report total number and show the sequences of the individual clones and the same clonal family per individual (using their definition of same VJ same CDR3 length and 80% homology of the CDR3). They should report how they determined antigen-specificity (if by generating monoclonal antibodies). Finally, they also report that these are public clones (figure 3c, d) and so they need to show the alignment of the VDJ sequences from different individuals to demonstrate the “public clone”. Finally, from this list, how many were private clones (not shared among individuals) and how many were public clones (shared between individuals). The alignments should be shown to illustrate public clones which are shared among individuals. If this were true that they found 749 public clonotypes, this would be a remarkable finding and needs to be shown clearly. The data of public RBD public clones are listed in the dataset 434723_1_data_set_8110896_s1l1j tab putative ASC_putative_RBD_spike_clones. In the dataset, there are very few that are shared among the individuals.

This information was previously requested to understand their datasets but they only change the text in the manuscript page line 191 to say that the 84 and 120 clones listed are “shared between individuals”. More data and clarification are required.

The 749 spike-specific public ASC clones from peripheral blood were identified according to the similarity of their Ig-sequences to sequences of spike-binding memory B cells (same sequences used for identification of spike-specific BMPC public clones). Figures 3c and d refer to these ASC clones. We have added reorganized tables now showing the aligning of the public/shared clones (see page 32, lines 1452 to 1456). Figure 3e refers to all of the ASC isolated from peripheral blood at the indicated time points after vaccination/infection (55071 clones). See revised version of the manuscript (page 5, lines 233 to 236).

4. (previous point #5) The authors themselves report that the blood ASC and BMPC transcriptomes do not overlap when they are evaluated by simple UMAP analysis. This statement needs to be included in the paper because it is misleading to conclude that overlaying the blood ASC onto the BMPC transcriptomes using their integrated analysis proves that these are the same cell transcriptionally.

They report that the UMAP analysis shows that they are very different and do not overlap; thus they cannot conclude that they are the same cells. However, they can report that they tried to suggest where these blood ASC may eventually reside in the BMPC by “using our integrated approach with anchor genes” in order to overlay the blood ASC onto the BMPC. Transcriptional analysis merely suggests and is hypothesis driving but they do not prove that certain blood ASC eventually reside in a particular BMPC compartment. For example, they do not time-stamp these cells or label them and follow them into the BMPC compartments. Thus, their conclusions are speculative and should be reported as such. This is human data and some experiments are limiting but there is still value in reporting as such but the interpretation should be reported appropriately so they do not make overarching conclusions from the dataset that cannot be proven.

They also say that blood ASC are different essentially due to “proliferation and mobility” genes and so they should remove the “proliferation and mobility” genes and show that they contain the similar transcriptomes as the BMPC if this is the only difference. This additional analysis would be highly welcome.

We have shown the integration and now also include the just aggregated UMAP in Supplementary Figure 3e. We also clarify the corrections used during the integration pipeline in the manuscript text (page 6, line 251). We do not state that blood ASC and BMPC are the same cell type, but that they share transcriptional imprinting.

5. (previous point #7). It is hard to conclude that primary immunization fills the BMPC clan 1 (CD19lo) and clan 0 (CD19hi) and secondary immunization fills the CD19hi or (clan 0). Interestingly, spike fills the CD19lo and CD19hi subsets with primary COVID-19 immunization but not in all patients as pointed out with subjects 555 and 556. Perhaps reporting that only 6/8 patients reported were demonstrate this finding? But now, the authors report that CD19lo subsets include clans 1, 4, 5, 6 to show demonstrate their point. If this is the case, why was this information not described previously. Now CD19hi clans now include 0, 8, 9, 11, 12, 13 and CD19lo included clans 1, 4, 5, and 6. If this is the case, they need to include this in the beginning of the description of the different clans. They probably need to expand supplementary figure 1e to include all the clans in the CD19hi and CD19lo groups.

As we had stated in the previous revision (point 7), we consider our conclusions to be conclusive. The fact that the different clans are either CD19high or CD19low had from the start been demonstrated in Figure 1c, 1d, Supplementary Figure 1e and especially 2d (see also page 4, lines 170 to 173). We highlight in the clan description the ones for which that difference is more accentuated, clan 0, clan 1 and clan 4 (page 3 line 119 to 124), also mentioning clan 5 (page 3 lines 126 and 127). Expanding Supplementary Figure 1e does not confer additional information. That primary immunization targets both CD19high and CD19low clans and secondary immunization preferentially the CD19high clans, at the time points analysed, is evident from the data in Figure 3. Supplementary Figure 2e shows that even in the small bone marrow samples obtained, public putatively spike-specific clones locate both to CD19high and CD19low compartments.

6. They conclude that their dataset shows that primary responses fill CD19lo and CD19hi while secondary responses only fill the CD19hi compartment. However, looking in the BMPC primary and secondary would look the same because one could not distinguish the pre-existing old PC (which should be in both CD19lo and CD19hi) vs new arrivals into the BMPC (CD19hi) by antigen-specificity alone. Thus, it is unclear how they can make this conclusion. Also, the BM samples were obtained from individuals with COVID vaccines 3 times and unknown in two patients of primary vs secondary responses. A single BMPC with antigen-specificity cannot separate old existing PC vs new arrivals in the BMPC with antigen-specificity alone. However, the authors are using their integrated approaches with the blood and BM ASC transcriptome overlay to draw this conclusion. The problem is that the blood ASC and BMPC transcriptomes do not overlap on the UMAP. Their integrative approach is speculative,

interesting, and intriguing but not conclusive that repeat priming does not replenish the CD19^{lo} compartments. They can only hypothesize and require further experiments for testing. Finally, in new figure after 3 doses of the vaccine (secondary response), they report higher numbers of CD19^{lo} antigen-specific ASC which is not consistent with their model. If with more 3 doses of the vaccine and increased time, how do they explain higher numbers of CD19^{lo} BMPC? Additional clarity is needed to interpret this dataset.

We state that our data show at the time points analysed, in secondary immune reactions, a clear predominance of generation of CD19^{high} ASC, generated in IL21-independent, presumably extrafollicular B cell (re)activations. This does not preclude subsequent generation also of CD19^{low} ASC in IL21-dependent, presumably follicular immune reactions, in secondary immune reactions. So over time, also the numbers of CD19^{low} BMPC would increase - see page 8 lines 350 to 356: "this increase is in line with the notion that secondary (and tertiary) immune reactions start with the extrafollicular reactivation of memory B cells, followed by generation of new and adapted ASC in a subsequent follicular immune reaction" (Alsoussi, W. B. *et al.* Nature 617, 592-598 (2023). <https://doi.org:10.1038/s41586-023-06025-4>).

REVIEWER COMMENTS

Reviewer #3 (Remarks to the Author):

I appreciate the additional data provided by the authors which provide deeper insights. Upon careful review of the “Sequence Alignment TT” and Sequence Alignment RBD spike” of the public clonotypes, the sequence alignments demonstrate major concerns about the definition of clones that are shared between two cells: particularly the antigen-specific B cell and BMPC clones. They say in line 187 “Based on CDR3 similarity, we designated as “public” RBD/spike and TT-specific clones, those ASC and BMPC clones with more than 80% similarity (modelled optimum for CDR3 specificity³¹) shared between individuals. Of the 38235 BMPC analyzed, 84 expressed spike-specific and 120 tetanus-specific public clonotypes (Figure 2e)”. Unfortunately, they do not show the V and J sequence, and so this cannot be evaluated. However, in the alignments provided, all clones appear to have the same CDR3 length but they do not have 80% homology and thus cannot be called a shared clone. One example of spike is BMPC clone in line 140 (barcode GACAGAGGTGATGTGG-16), which does not align with any BCR-spike-specific B cell or any other sequence grouped in this clone (the most similar heavy chain CDR3 in this grouped clone is 70.83%, which is well below the 80% threshold). Many other HCDR3s in this first grouping have similarities as low as 29% relatedness with other sequences. Similar patterns occur through all sequence alignments of spike and TT-specific clones. Another example in the TT-specific BMPC clones occurs in line 24 and 33, which do not share alignment with a BCR-TT-specific B cell. Hence, the authors call these spike and TT-specific BMPC clones based on the sequences alignments which are not correct.

To provide further detail, from the 120 putative TT-specific clonotypes, many of the CDR3 sequences from the putative TT-specific BMPC do not seem to align with a TT-specific B cell (See REANAL_Sequence_Alignment_RBD/S). They have the same CDR3 length but not the 80% homology. If the antigen-specific BCR sequences from memory B cell do not align with the same clonotype as the BMPC, they are not likely TT- or RBD-specific BMPC. The BMPC TT-clones only align by the same CDR3 length but not always by 80% CDR3 homology with a TT-specific B cell CDR3. Almost always, this sharing comes from the BMPC and BCR TT-specific B cell from the same individual. Of the 120 clones, 70 appear to align with a TT-

specific B cell sequence and I agree likely to be TT-specific BMPC. The remaining 50 BMPC of 120 are not likely to be TT-specific because they do not have 80% CDR3 homology with any TT-specific B cell. Oftentimes, these BMPC clones and TT-specific B cell clones that do not align often come from the different individuals. Thus, not all original 120 clones as assigned TT-specific in the BMPC is necessarily TT specific. This data would need to be reanalyzed to exclude the BMPC clones without matching TT-specific BCR CDR3 sequences by length and 80% homology, and V/J gene information should be used to further specify clonal clustering. (Please focus on column FS and FT and with summary in FW of excel file REANAL_Sequence_Alignment_RBD/S)

For the putative spike-specific BMPC based on the same definition of a shared clone, CDR3 length but 80% homology between a spike-specific B cell in alignment, of the 84 putative spike-specific clones, only 45 are spike-specific BMPC. 39 are not likely spike-specific clones since the CDR3 does not have 80% homology. (see files REANAL_Sequence_alignment_TT, please focus on column F and G with summary in AQ).

In all, it appears it is 50 /120 (41.7%) of the putative TT-specific BMPC are not likely TT-specific BMPC and the 39/84 (46.4%) of the putative spike-specific BMPC are not likely spike-specific BMPC. This is highly problematic since locating antigen-specific BMPC based on the VDJ sequences was the critical element of the paper. Since clonality appears to be assigned by CDR3 length but not 80% homology of the CDR3 of the clones, this approach resulted in many false positive alignments. We could not follow the same V and J because this information was not provided. Since they did not show antigen-specificity from all the clones by monoclonal antibody generation, the sequence data is absolutely critical.

Unfortunately, the alignments provided are not convincing that all assigned TT-specific and spike-specific BMPC clones are indeed antigen-specific. Thus, it is very difficult to make conclusions of the antigen-specific BMPC in the single cell UMAP. (Figures 2 and 3 would need re-analysis).

As to calling them all “public clonotypes”, for tetanus, there appears to be 30 clones but only 11 of the 30 clones showed sequences from different individuals. 19 of the 30 clones contained sequences only from the same individual which would exclude them from the

definition of a public clone. There also appeared to be only 8 sequences that would be considered a public clonotype based on the definition of 80% CDR3 homology from different donors provided in line 187 of the manuscript. Therefore, it appears inaccurate to call all of these public clonotypes since the majority of the clones are not shared between individuals.

Response to Reviewer 3:

Reviewer 3 states: I appreciate the additional data provided by the authors which provide deeper insights. Upon careful review of the “Sequence Alignment TT” and Sequence Alignment RBD spike” of the public clonotypes, the sequence alignments demonstrate major concerns about the definition of clones that are shared between two cells: particularly the antigen-specific B cell and BMPC clones . They say in line 187 “Based on CDR3 similarity, we designated as “public” RBD/spike and TT-specific clones, those ASC and BMPC clones with more than 80% similarity (modelled optimum for CDR3 specificity³¹) shared between individuals. Of the 38235 BMPC analyzed, 84 expressed spike-specific and 120 tetanus-specific public clonotypes (Figure 2e)”. Unfortunately, they do not show the V and J sequence, and so this cannot be evaluated. However, in the alignments provided, all clones appear to have the same CDR3 length but they do not have 80% homology and thus cannot be called a shared clone. One example of spike is BMPC clone in line 140 (barcode GACAGAGGTGATGTGG-16), which does not align with any BCR-spike-specific B cell or any other sequence grouped in this clone (the most similar heavy chain CDR3 in this grouped clone is 70.83%, which is well below the 80% threshold). Many other HCDR3s in this first grouping have similarities as low as 29% relatedness with other sequences. Similar patterns occur through all sequence alignments of spike and TT-specific clones. Another example in the TT-specific BMPC clones occurs in line 24 and 33, which do not share alignment with a BCR-TT-specific B cell. Hence, the authors call these spike and TT-specific BMPC clones based on the sequences alignments which are not correct.

To provide further detail, from the 120 putative TT-specific clonotypes, many of the CDR3 sequences from the putative TT-specific BMPC do not seem to align with a TT-specific B cell (See REANAL_Sequence_Alignment_RBD/S). They have the same CDR3 length but not the 80% homology. If the antigen-specific BCR sequences from memory B cell do not align with the same clonotype as the BMPC, they are not likely TT- or RBD-specific BMPC. The BMPC TT-clones only align by the same CDR3 length but not always by 80% CDR3 homology with a TT-specific B cell CDR3. Almost always, this sharing comes from the BMPC and BCR TT-specific B cell from the same individual. Of the 120 clones, 70 appear to align with a TT-specific B cell sequence and I agree likely to be TT-specific BMPC. The remaining 50 BMPC of 120 are not likely to be TT-specific because they do not have 80% CDR3 homology with any TT-specific B cell. Oftentimes, these BMPC clones and TT-specific B cell clones that do not align often come from the different individuals. Thus, not all original 120 clones as assigned TT-specific in the BMPC is necessarily TT specific. This data would need to be reanalyzed to exclude the BMPC clones without matching TT-specific BCR CDR3 sequences by length and 80% homology, and V/J gene information should be used to further specify clonal clustering. (Please focus on column FS and FT and with summary in FW of excel file REANAL_Sequence_Alignment_RBD/S)

For the putative spike-specific BMPC based on the same definition of a shared clone, CDR3 length but 80% homology between a spike-specific B cell in alignment, of the 84 putative spike-specific clones, only 45 are spike-specific BMPC. 39 are not likely spike-specific clones since the CDR3 does not have 80% homology. (see files REANAL_Sequence_alignment_TT, please focus on column F and G with summary in AQ).

In all, it appears it is 50 /120 (41.7%) of the putative TT-specific BMPC are not likely TT-specific BMPC and the 39/84 (46.4%) of the putative spike-specific BMPC are not likely spike-specific BMPC. This is highly problematic since locating antigen-specific BMPC based on the VDJ sequences was the critical element of the paper. Since clonality appears to be assigned by CDR3 length but not 80% homology of the CDR3 of the clones, this approach resulted in many false positive alignments. We could not follow the same V and J because this information was not provided. Since they did not show antigen-specificity from all the clones by monoclonal antibody generation, the sequence data is absolutely critical. Unfortunately, the alignments provided are not convincing that all assigned TT-specific and spike-specific BMPC clones are indeed antigen-specific. Thus, it is very difficult to make conclusions of the antigen-specific BMPC in the single cell UMAP. (Figures 2 and 3 would need re-analysis).

As to calling them all “public clonotypes”, for tetanus, there appears to be 30 clones but only 11 of the 30 clones showed sequences from different individuals. 19 of the 30 clones contained sequences only from the same individual which would exclude them from the definition of a public clone. There also appeared to be only 8 sequences that would be considered a public clonotype based on the definition of 80% CDR3 homology from different donors provided in line 187 of the manuscript. Therefore, it appears inaccurate to call all of these public clonotypes since the majority of the clones are not shared between individuals.

Response:

We agree with the reviewer that the definition of “antigen-specific B cell clone” in this manuscript is based on sequence comparisons and not on actual binding analyses of plasma cell antibodies. The latter would have been outside the scope of the manuscript. The definition of acceptable homology ranges is based on the publication of Bürckert et al. ref. 31, as cited in the manuscript (ref. 31). In this publication, more than 80% homology of heavy chains ONLY is shown to be the optimal range for acceptable somatic hypermutation and maintenance of antigen-specificity. Data on light chains is not included. With respect to our data, in which sequences of heavy and light chains are available, asking for more than 80% of homology of ONLY the heavy chains would have defined more “clones” than asking for more than 80% of homology of heavy and light chains combined (see table 1).

Criteria	# of cells	
	heavy chain + light chain homology > 80%	heavy chain homology > 80%
BMPC_putative_TT_clones	120	187
ASC_putative_RBD_spike_clones	749	1176
BMPC_putative_RBD_spike_clones	84	253

Table 1. Number of BMPC or ASC identified as expressing TT- or spike-specific antibodies based on different criteria to define antigen specific clones.

The suggestion to be even more restrictive, by asking for more than 80% homology of heavy and light chains combined AND additionally for more than 80% of homology of the heavy chain alone, does not seem to be based on any scientific evidence. We do not see a reason for this additional restriction, nor for a recalculation of “clones” according to this ad hoc definition. We would like to stay with our definition, which now is even more clearly

explained in the revised version of the manuscript (page 4, line 187, page 15 lines 710 to 718 and pages 32 and 33 lines 1470 to 1495).

This reviewer also questions our definition of “public” clones, notifying that some clones, as we define them, are predominantly expressed in one individual. Again, we would like to maintain our clear definition that any clone present in more than one individual is a “public” clone.

REVIEWERS' COMMENTS

Although helpful, using the light and heavy chain is not absolutely necessary to define a clone. However, in their analyses, using only the heavy chain, their alignment of the BCR and BMPC sequences is inaccurate.

For clarity, in my analyses, I used the author's definition of a clone with the same CDR3 length and 80% homology of the CDR3 for the extensive re-analysis of the data which took a considerable amount of my time to make certain of the results. Using the author's data sequences and the author's definition of a clone ((same CDR3 length and 80% homology), I aligned the tetanus-BCR memory clones with the BMPC clones or aligned the spike-specific BCR memory clones with the BMPC clones. Using the author's datasets and the author's definition of a clone only 70/120 TT-specific BMPC align with a TT-memory B cell clone while 50/120 (41%) TT-specific BMPC clones DO NOT align with a TT-memory B cell clone. Thus, 50 could not be called TT-specific BMPC. Also, only 45/84 spike-specific BMPC aligned. 39/84 (46%) of the spike-specific BMPC clones DO NOT align with the spike specific-memory B cell clones. Thus, they cannot be called spike-specific BMPC. Therefore, many figures needed to be re-analyzed (especially figure 2 and supplemental figure 2). I had hoped that they would have reanalyzed the data with these findings to see where the spike-specific and TT-specific clones resided in the BMPC compartment. They seem to argue that they used 80% homology which I also used in my re-analysis of the data even though the field may use 85% homology of the CDR3. Nonetheless, using the author's definitions to define a matching clone, I believe there are many clones which are not TT and not spike specific in the BMPC which make conclusions inaccurate. Based on these findings, I am not able to approve this manuscript for publication.

POINT-BY-POINT RESPONSE TO REVIEWERS' COMMENTS

Although helpful, using the light and heavy chain is not absolutely necessary to define a clone. However, in their analyses, using only the heavy chain, their alignment of the BCR and BMPC sequences is inaccurate.

For clarity, in my analyses, I used the author's definition of a clone with the same CDR3 length and 80% homology of the CDR3 for the extensive re-analysis of the data which took a considerable amount of my time to make certain of the results. Using the author's data sequences and the author's definition of a clone ((same CDR3 length and 80% homology), I aligned the tetanus-BCR memory clones with the BMPC clones or aligned the spike-specific BCR memory clones with the BMPC clones. Using the author's datasets and the author's definition of a clone only 70/120 TT-specific BMPC align with a TT-memory B cell clone while 50/120 (41%) TT-specific BMPC clones DO NOT align with a TT-memory B cell clone. Thus, 50 could not be called TT-specific BMPC. Also, only 45/84 spike-specific BMPC aligned. 39/84 (46%) of the spike-specific BMPC clones DO NOT align with the spike specific-memory B cell clones. Thus, they cannot be called spike-specific BMPC. Therefore, many figures needed to be re-analyzed (especially figure 2 and supplemental figure 2). I had hoped that they would have reanalyzed the data with these finding to see where the spike-specific and TT-specific clones resided in the BMPC compartment. They seem to argue that they used 80% homology which I also used in my re-analysis of the data even though the field may use 85% homology of the CDR3. Nonetheless, using the author's definitions to define a matching clone, I believe there are many clones which are not TT and not spike specific in the BMPC which make conclusions inaccurate. Based on these findings, I am not able to approve this manuscript for publication.

Using our definition for public clone, which means that a BCR sequence of a BMPC exhibits over 80% CDR3 sequence identity in both heavy and light chains when compared to the BCR of a sequenced experimentally validated spike- or tetanus-specific cell, we could indeed identify 120 TT-specific BMPC and 84 spike-specific BMPC. All corresponding alignments are shown in the Supplementary Data file. We believe the discrepancy between ours and the reviewer's calculations comes from the fact that the reviewer has only taken the heavy chain into consideration, and therefore only identified a fraction of our clones as specific.

If we had initially assessed only the heavy chain sequences for over 80% identity within the entire dataset and not our more stringent criteria, we would have identified even more sequences, as indicated in Table 1. It is important to note that the entire RNA sequencing dataset, which is accessible on the GEO repository under accession number GSE253862, goes beyond what has been accessible to the reviewers in our Supplementary Data file. There, only clones filtered to ensure they had over 80% sequence identity, considering both heavy and light chain sequences of the B-cell receptors (BCRs) combined, were displayed.

The 80% homology criterion was chosen based on a publication, cited in our manuscript, showing 80% to be the modelled optimum for CDR3 specificity. Therefore, we are confident our conclusions are accurate.

Criteria	# of cells	
	heavy chain + light chain homology > 80%	heavy chain homology > 80%
BMPC_putative_TT_clones	120	187
ASC_putative_RBD_spike_clones	749	1176
BMPC_putative_RBD_spike_clones	84	253

Table 1. Number of BMPC or ASC identified as expressing TT- or spike-specific antibodies based on different criteria to define antigen specific clones.